# Domain-Specific Pruning of Large Mixture-of-Experts Models with Few-shot Demonstrations

**Zican Dong**[1,2]\*, **Han Peng**[1,2]\*, **Peiyu Liu**[3]†, **Wayne Xin Zhao**[1,2]†,
**Dong Wu**[5], **Feng Xiao**[4], **Zhifeng Wang**[4]
[1] Gaoling School of Artificial Intelligence, Renmin University of China
[2] Beijing Key Laboratory of Research on Large Models and Intelligent Governance
[3] University of International Business and Economics
[4] YanTron Technology Co. Ltd    [5] EBTech Co. Ltd
{dongzican, panospeng}@ruc.edu.cn
liupeiyustu@163.com, batmanfly@gmail.com,
wudong@yantronic.com, {fengx, zhifengw}@ebtech.com

## Abstract

Mixture-of-Experts (MoE) models achieve a favorable trade-off between performance and inference efficiency by activating only a subset of experts. However, the memory overhead of storing all experts remains a major limitation, especially in large-scale MoE models such as DeepSeek-R1 (671B). In this study, we investigate domain specialization and expert redundancy in large-scale MoE models and uncover a consistent behavior we term *few-shot expert localization*, with only a few in-domain demonstrations, the model consistently activates a sparse and stable subset of experts on tasks within the same domain. Building on this observation, we propose a simple yet effective pruning framework, **EASY-EP**, that leverages a few domain-specific demonstrations to identify and retain only the most relevant experts. EASY-EP comprises two key components: **output-aware expert importance assessment** and **expert-level token contribution estimation**. The former evaluates the importance of each expert for the current token by considering the gating scores and L2 norm of the outputs of activated experts, while the latter assesses the contribution of tokens based on representation similarities before and after routed experts. Experiments on DeepSeek-R1 and DeepSeek-V3-0324 show that our method can achieve comparable performances and $2.99\times$ throughput under the same memory budget as the full model, with only half the experts. Our code is available at https://github.com/RUCAIBox/EASYEP.

## 1 Introduction

Mixture-of-Experts (MoE) architectures have been widely adopted as the backbones of various large language models (LLMs) due to their efficiency of scaling parameters without proportional computational overhead [1–4]. However, the deployment of large MoE models imposes substantial memory requirements. Taking DeepSeek-R1 (671B) [1] as an example, it takes about 1500 GB under BF16 precision and 750 GB under FP8 precision, necessitating 4×8 A800 or 2×8 H800 GPU configurations, respectively. This underscores the critical need to explore lite deployment strategies for large-scale MoE models like DeepSeek-R1.

Various training-free approaches have been proposed to alleviate the inference memory demands of MoE models. Expert pruning reduces memory by removing less important experts. Among them,

---

\*Equal Contribution.
†Corresponding author.

39th Conference on Neural Information Processing Systems (NeurIPS 2025).

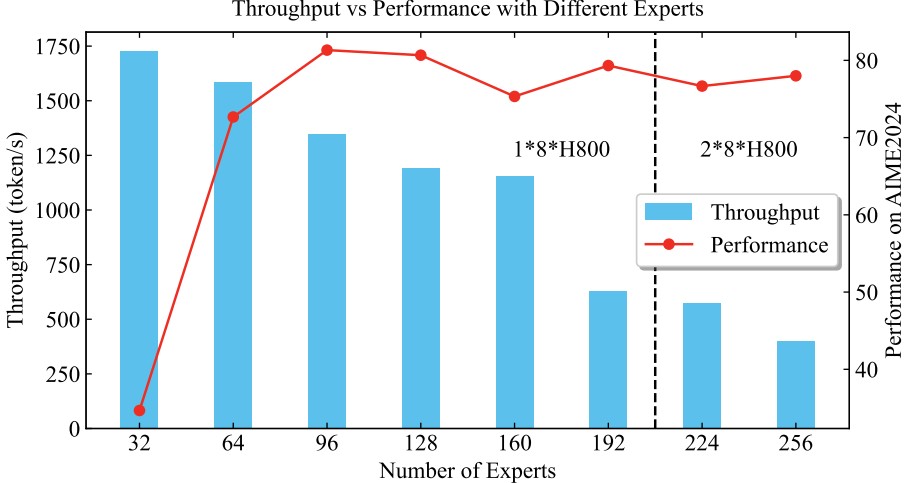

Figure 1: Throughput and performance comparison of DeepSeek-R1 on AIME2024 with varying expert numbers using EASY-EP. We deploy DeepSeek-R1 with two $8\times$ H800 for 224 and 256 experts, while one $8\times$ H800 for others. The throughputs of the latter configurations are multiplied by 2.

router-based methods use expert activation statistics to estimate importance [5], while perturbation-based ones select expert subsets to minimize hidden state drift [6, 7]. However, the former may fail to identify key experts, and the latter is expensive as the number of experts increases. Expert merging combines similar experts to reduce their number [8, 9], but can cause neuron misalignment of experts in MoE models trained from scratch [10, 11]. Moreover, existing methods are primarily designed for MoEs with a few experts per layer (*e.g.,* Mixtral $8\times$7B [2]) [6]. This highlights the need for more scalable and accurate pruning strategies tailored to large-scale MoE models. While such a scale poses challenges for current approaches, it also brings new opportunities: the increased granularity and domain specialization of experts in models like DeepSeek-R1 [12, 13] make them especially amenable to domain-specific pruning at high compression ratios [14].

In this study, we analyze how expert activations in a large MoE model vary across domains and respond to a small number of demonstration samples. We observe a consistent behavior: given just a few domain-specific examples, the model tends to activate a stable and sparse subset of experts that are highly relevant to the target domain. We refer to this phenomenon as *few-shot expert localization*. To better understand the mechanism behind this behavior, we focus on two factors: the domain specialization of experts and the sufficiency of limited demonstrations. Our key findings are as follows: (1) High gating-value experts are strongly domain-specific, consistently dominating activations within their respective domain while remaining inactive in unrelated ones. (2) A small number of demonstrations suffices to reliably trigger these experts, and they generalize well to other unseen datasets within the same domain.

Based on our observations, we introduce a domain-specific pruning framework that leverages few-shot demonstrations to address memory constraints in large-scale MoE models. Our approach begins by sampling a small number of task demonstrations from a specific domain and generating responses with the original model, serving as a calibration set. To identify and retain the most critical experts, we then develop a pruning method, **E**xpert **A**ssessment with **S**imple **Y**et-effective scoring for **E**xpert **P**runing, *a.k.a.,* **EASY-EP**. This method estimates expert importance and retain a fixed-size subset of top-scoring experts. EASY-EP consists of two complementary components: (1) output-aware expert importance assessment, which combines gating values and the L2 norm of expert outputs to estimate per-token expert importance, and (2) expert-level token contribution estimation, which measures the similarity between the input and the residual-connected output of routed experts to measure each token's contribution to the overall expert score. Notably, the entire scoring process requires only a single forward pass, eliminating the need for backpropagation or repeated evaluations.

To evaluate the efficacy of our approach, we conducted systematic experiments on DeepSeek-R1 and DeepSeek-V3-0324 using eight benchmark datasets covering math, coding, science, finance,

medicine, and agent execution capabilities. Specifically, with only retaining $50\%$ experts, our method can keep comparable performances under the domain-specific pruning settings while achieving over $90\%$ of the full model's performances on DeepSeek-R1 and even better performances on DeepSeek-V3-0324 under the mixed-domain pruning settings. As shown in Figure 1, the performances degrade only slowly with the increase of compression ratio, indicating robustness to pruning. Additionally, under identical memory constraints, pruning $50\%$ of the experts yields a $2.99\times$ increase in inference throughput for sequences of 1K input and 1K output lengths, highlighting the practical utility of our framework in real-world deployments.

## 2   Background

MoE architectures introduce MoE modules where parameters are dynamically activated to replace feedforward networks (FFNs) [2, 1]. Specially, in the $l$-th layer, a MoE module contains a router $G(\cdot)$ and $N$ routed experts $\{E_1^l(\cdot), \ldots, E_N^l(\cdot)\}$ [3]. Given an input representation sequence $\mathbf{H}^l = \{\boldsymbol{h}_1^l, \ldots, \boldsymbol{h}_T^l\}, \forall \boldsymbol{h}_t^l \in \mathbb{R}^D$, the router computes the logit of each expert for the $t$-th token and applies a gating function on the Top-$K$ logits to obtain the gating values $g_{i,t}^l$ (the gating values of deactivated experts $g_{i,t} = 0$). The Top-$K$ experts are activated, and their outputs are aggregated via weighted summation. The final output is obtained by residual connections of the input and output of experts:

$$\bar{\boldsymbol{h}}_t^l = \sum_{i=1}^{N} g_{i,t}^l \cdot E_i^l(\boldsymbol{h}_t^l), \quad \tilde{\boldsymbol{h}}_t^l = \boldsymbol{h}_t^l + \bar{\boldsymbol{h}}_t^l \tag{1}$$

With the gating values of the router, we define two metrics to assess the importance of each expert, *i.e., frequency* and *gating scores* [5]. For all tokens in a calibration set and an expert $E_i^l$, we define the frequency $f_i^l$ as the number of times each expert is activated, while the gating scores $r_i^l$ is defined as the total sum of the gating values when each expert is activated, as shown in Equation 2. Here, $M$ is the size of the calibration set, and $T_n$ denotes the number of tokens per demonstration.

$$f_i^l = \sum_{n=1}^{M} \sum_{t=1}^{T_n} (g_{i,n,t}^l > 0), \quad r_i^l = \sum_{n=1}^{M} \sum_{t=1}^{T_n} g_{i,n,t}^l, \tag{2}$$

## 3   Empirical Analysis of Experts

Previous work has demonstrated that the expert distributions of MoEs with few experts (*e.g.,* Mixtral $8 \times 7B$) mainly depend on the syntax structures instead of domains [2]. However, recent large-size MoE models are equipped with various fine-grained experts (*e.g.,* DeepSeek-R1 has 256 experts per layer), which may be more specialized and store distinct knowledge and capacities in their parameters. Motivated by this, we empirically study a phenomenon we term *few-shot expert localization*, where domain-specific experts can be reliably identified using only a handful of demonstrations.

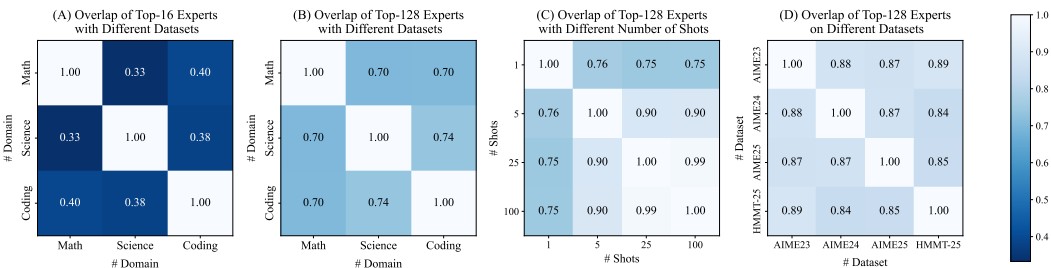

Figure 2: (A), (B): Overlap ratios of experts with top gating scores on different datasets. (C): Overlap ratio of top-128 experts with different numbers of demonstrations. (D): Overlap ratio of top-128 experts pruned with different math datasets.

---

[3]Shared experts are employed in some MoE models [1, 14–16], but are not considered in this work.

Table 1: Results of removing domain-specific experts. **Bold** denotes in-domain results.

| Domain | AIME24 | GPQA | LiveCodeBench |
|---|---|---|---|
| Full | 77.08 | 70.91 | 63.32 |
| Math | 67.33 **(-9.75)** | 69.19 (-1.72) | 65.27 (+1.95) |
| Code | 78.67 (+1.59) | 71.72 (+0.81) | **55.68 (-6.64)** |
| Science | 79.33 (+2.25) | **59.09 (-11.82)** | 61.07 (-2.25) |

## 3.1 Expert Specialization Across Domains

To assess whether experts in large MoE models exhibit domain-specific specialization, we select AIME-2023 [17], GPQA-main [18], and LiveCodeBench-V3 [19] as calibration datasets for the domains of math, science, and coding, respectively. We conduct experiments using the representative model DeepSeek-R1 on these datasets and extract the gating scores across different domains (as described in Section 2). By analyzing the expert activation distributions across these domains, we aim to reveal how expert utilization varies and assess the degree of specialization.

**Distinct Expert Distribution Across Domains.** We first rank all experts at each layer by their gating scores and select the top-16 and top-128 experts for each dataset. Subsequently, we measure the overlap of top-ranked experts across different domains and visualize the overlap ratios in Figure 2, where (A) corresponds to top-16 and (B) to top-128. We observe that the top-16 experts are largely disjoint across datasets. When expanding the selection to top-128, the degree of overlap increases, but a significant portion of the experts remains domain-specific. This indicates that *large MoE models contain domain-specialized experts that are predominantly activated in their respective domains.* We show experiments on different numbers of top experts and layer variations in Appendix A.

**Impact of Removing Domain-Specific Experts.** In order to explore the importance of domain-specific experts, we remove those that appear in the top-128 (by gating score) in a specific domain but not in any others. We then evaluate each pruned model on tasks from the same domain as the calibration data (*i.e.,* in-domain), and on tasks from other domains (*i.e.,* out-of-domain), to assess the generalization behavior of the remaining experts. As shown in Table 1, pruning these experts leads to significant performance degradation on in-domain tasks while having minimal impact on out-of-domain tasks. These results suggest that *domain-specific experts play a critical role in the relevant domain but are redundant for other domains.*

## 3.2 Expert Locality Within One Domain

Beyond examining the expert specialization across different domains, we examine the locality and stability of expert activation within a single domain. We investigate how the number of demonstrations and the choice of calibration set influence expert selection patterns under the same domain setting.

**Effect of Calibration Set Size.** Given the presence of domain-specific experts in DeepSeek-R1, a fundamental question arises: How many demonstrations are necessary to accurately identify these key experts? To answer this, we sample varying numbers of demonstrations (*i.e.,* 1, 5, 25, 100) from LiveCodeBench-v3 [19]. For each setting, we compute the average gating scores and retain the top 128 experts. Based on these selections, we then calculate the pairwise overlap ratios between expert sets derived from different demonstration sizes. As illustrated in Figure 2 (C), even with just five demonstrations, over 90% of the critical experts can be effectively identified. Furthermore, 25 demonstrations are sufficient to capture all domain-specific experts (99%), with additional demonstrations yielding only marginal improvements. These results underscore *the feasibility of few-shot domain-specific expert pruning.*

**Consistency of Expert Activation Across Datasets.** To investigate the consistency of domain-specific experts across datasets within the same domain, we conduct experiments with DeepSeek-R1 on four math datasets: AIME-2023, AIME-2024, AIME-2025, and HMMT-Feb 2025 [17]. Specifically, we perform expert pruning under the 25-shot setting for each dataset and retain Top-128 experts based on their average gating scores. We then compute the pairwise overlap ratios between each pair of datasets. As shown in Figure 2 (D), the overlaps exceed 84% across all math datasets,

revealing a high degree of consistency in domain-specific expert activation. This indicates that *domain-specific expert activation patterns are largely transferable* within the same domain. Despite some dataset-specific differences, the overall expert overlap remains strong and stable.

Empirical analysis with another metric is shown in Appendix B.

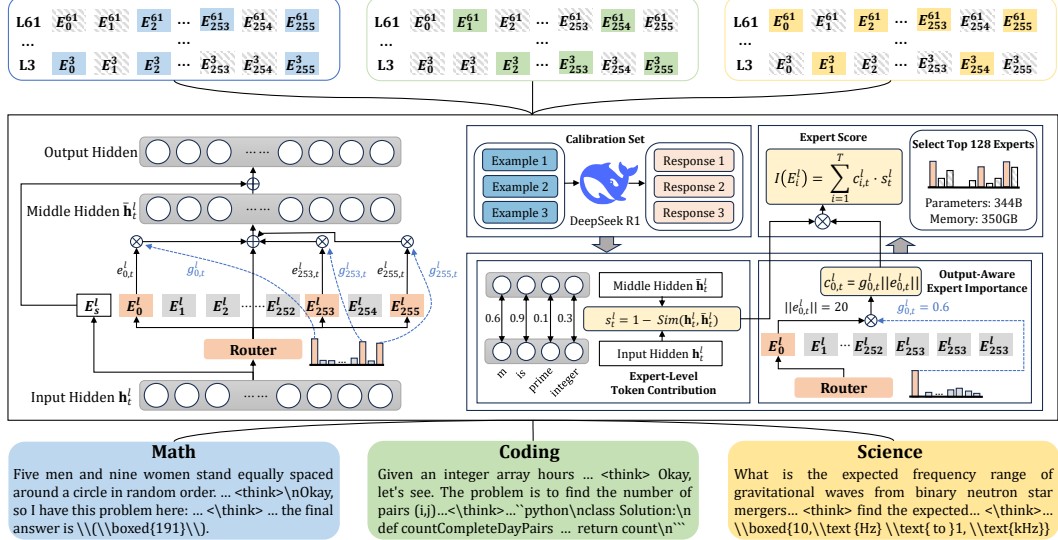

Figure 3: Overall framework of EASY-EP. Given a calibration set consisting of input and responses by the model, EASY-EP leverages output-aware expert importance assessment and expert-level token contribution estimation to compute the expert score on the domain and returns the pruned expert sets.

## 4 Method

### 4.1 Overview

Motivated by our earlier observation of few-shot expert localization phenomena in Section 3, we propose an expert pruning framework to reduce memory costs. We use a small set of target-domain demonstrations to run the MoE model and collect expert activation statistics, considering both inputs and outputs for pruning. To effectively identify domain-specific experts in large MoE models, we introduce a simple yet effective expert pruning method, **EASY-EP**. Specifically, we first compute the product of the expert output L2 norm and its corresponding gating value as the output-aware expert importance $c_{i,t}^l$. Next, we determine the expert-level token contribution $s_t^l$ based on the similarity of representations before and after expert computation. The final expert score $I(\mathrm{E}_i^l)$ is obtained by aggregating the product of two terms over all tokens:

$$I(\mathrm{E}_i^l) = \sum_{t=1}^{T} c_{i,t}^l \cdot s_t^l. \tag{3}$$

Our method also supports mixed-domain pruning by averaging the normalized expert scores of all target domains. Thus, we can prune a single model to handle tasks across multiple domains:

$$I_{mix}(\mathrm{E}_i^l) = \sum_{\tau \in \mathcal{T}} (I_\tau(\mathrm{E}_i^l) / \sum_{j=1}^{N} I_\tau(\mathrm{E}_j^l)). \tag{4}$$

Based on the expert scores computed from a small subset of data, we can efficiently select the Top-$M$ experts with the highest scores as retained experts while pruning other experts to reduce memory costs. The overall framework of our method is illustrated in Figure 3.

### 4.2 Output-Aware Expert Importance Assessment

To assess the importance of each expert, prior router-based expert pruning methods assume activated expert gating scores can reflect their importance [5]. However, this assumption has not considered

the influence of experts. To further assess the contribution of each routed expert, we example the aggregated output $\bar{\boldsymbol{h}}_t^l$ from all routed experts for a given token:

$$\bar{\boldsymbol{h}}_t^l = \sum_{i=1}^{N} g_{i,t}^l \cdot \boldsymbol{e}_{i,t}^l = \sum_{i=1}^{N} g_{i,t}^l \|\boldsymbol{e}_{i,t}^l\| \cdot \frac{\boldsymbol{e}_{i,t}^l}{\|\boldsymbol{e}_{i,t}^l\|}, \tag{5}$$

where $\|\cdot\|$ denotes the L2 norm of a vector, $\boldsymbol{e}_{i,t}^l = \mathrm{E}_i^l(\boldsymbol{h}_t^l)$ represents the output of the expert, and $\frac{\boldsymbol{e}_{i,t}^l}{\|\boldsymbol{e}_{i,t}^l\|}$ denotes the unit vector in the direction of the expert's output. Further, we can compute the upper bound of the L2 norm of the expert outputs as follows:

$$\|\bar{\boldsymbol{h}}_t^l\| \leq \sum_{i=1}^{N} \left\| g_{i,t}^l \|\boldsymbol{e}_{i,t}^l\| \cdot \frac{\boldsymbol{e}_{i,t}^l}{\|\boldsymbol{e}_{i,t}^l\|} \right\| = \sum_{i=1}^{N} g_{i,t}^l \|\boldsymbol{e}_{i,t}^l\|. \tag{6}$$

This indicates that each expert's contribution to the final output is bounded by the product of its gating value and the L2 norm of its output, $g_{i,t}^l \|\boldsymbol{e}_{i,t}^l\|$. An expert with a large gating score may still produce outputs with low L2 norm, ultimately resulting in a limited influence on the final output, which are empirically verified in Appendix C.1. Therefore, instead of using only the gating value, we define the importance of an expert for a given token as the product of its gating value and L2 norm of output, as formalized in the following equation:

$$c_{i,t}^l = g_{i,t}^l \|\boldsymbol{e}_{i,t}^l\|, \quad \forall g_{i,t}^l > 0. \tag{7}$$

### 4.3 Expert-Level Token Contribution Estimation

When calculating the statistical metrics for evaluating the importance of experts, prior work often directly averages the scores across all tokens [6, 5]. However, in practice, the influence of routed experts' outputs on the residual stream varies significantly across tokens (as shown in Appendix C.2). Intuitively, when dealing with tokens exhibiting low similarity before and after the MoE module, adjusting their routed experts will induce a substantial distributional shift in their representations. In contrast, for tokens with high similarity, such adjustments will lead to only minimal drift in their representational distributions [20, 21]. Inspired by these, we propose a similarity-based token importance assessment method which give greater weights for the former tokens. Given the representations before and after the routed expert modules $\boldsymbol{h}_t^l$ and $\tilde{\boldsymbol{h}}_t^l$, we compute the cosine similarity between these representations. The token importance score $s_t^l$ is then defined as one minus this similarity, capturing the extent of change induced by the routed expert module:

$$s_t^l = 1 - \mathrm{Sim}(\boldsymbol{h}_t^l, \tilde{\boldsymbol{h}}_t^l). \tag{8}$$

## 5 Experiments

### 5.1 Experimental Settings

**Evaluation Benchmarks.** To systematically assess the effectiveness of our proposed method, we conduct experiments across eight benchmark datasets: AIME-2024, AIME-2025, HMMT-Feb 2025, LiveCodeBench [19], GPQA-Diamond [18], USMLE [22], FinanceIQ [23], and AgentBench-OS [24]. These benchmarks encompass six fundamental domains and tasks of LLMs: math, coding, science, medicine, finance, and agent-based task execution.

**Experiment Settings.** We select DeepSeek-R1 [1] and DeepSeek-V3-0324 [15] as our evaluated models and consider domain-specific and mixed-domain pruning settings. For each domain, we randomly sample 25 instances and construct a calibration set by concatenating their inputs with the target model's outputs. We then evaluate the expert scores on the calibration data and select the top 64 and 128 experts with the highest scores at each layer, respectively. We also average the normalized expert scores on different domains to evaluate the mixed-domain pruning performances. Details regarding the candidate sets and evaluation settings are provided in Appendix D.

Table 2: Comparison of the performances of different expert pruning methods. HMMT denotes HMMT-Feb 2025, GPQA denotes GPQA-Diamond, A-OS denotes AgentBench-OS, and FinIQ denotes FinanceIQ.

| Model | Method | Mix | #E | AIME-24 | AIME-25 | FMMT | LiveCode | GPQA | USMLE | FinIQ | A-OS | Avg |
|---|---|---|---|---|---|---|---|---|---|---|---|---|
| | Full | - | 256 | 77.08 | 66.67 | 44.38 | 63.32 | 70.91 | 92.66 | 82.1 | 40.51 | 67.20 |
| | Random | × | 64 | 0.00 | 0.00 | 0.00 | 0.00 | 26.09 | 0.00 | 0.00 | 0.00 | 3.26 |
| | Frequency | × | 64 | 0.00 | 0.00 | 0.00 | 0.00 | 17.68 | 0.00 | 0.00 | 2.78 | 2.58 |
| | Gating Score | × | 64 | 2.67 | 1.33 | 2.67 | 14.97 | 46.83 | 0.86 | 0.00 | 0.69 | 8.75 |
| | M-SMoE | × | 64 | 0.00 | 0.00 | 0.00 | 0.00 | 12.12 | 0.00 | 0.00 | 0.00 | 1.52 |
| | EASY-EP | × | 64 | **72.81** | **55.10** | **38.02** | **42.51** | **67.47** | **26.63** | **33.90** | **27.26** | **45.22** |
| DeepSeek | Random | × | 128 | 8.33 | 6.67 | 3.33 | 20.96 | 34.95 | 57.66 | 0.00 | 7.64 | 17.44 |
| -R1 | Frequency | × | 128 | 19.33 | 13.33 | 7.33 | 36.08 | 59.60 | 61.51 | 26.40 | 29.16 | 31.59 |
| | Gating Score | × | 128 | 70.10 | 55.52 | 36.15 | 47.60 | 63.78 | 80.36 | 66.50 | 31.94 | 56.49 |
| | M-SMoE | × | 128 | 5.33 | 6.00 | 3.33 | 25.75 | 24.75 | 52.63 | 39.60 | 19.44 | 22.10 |
| | EASY-EP | × | 128 | **79.17** | **68.33** | **45.31** | **61.11** | **70.12** | **91.67** | **78.80** | **37.92** | **66.55** |
| | Frequency | ✓ | 128 | 21.33 | 10.00 | 6.00 | 7.49 | 41.45 | 78.55 | **62.14** | 11.81 | 29.85 |
| | Gating Score | ✓ | 128 | 29.33 | 21.33 | 18.00 | 22.75 | 41.69 | 62.06 | 27.29 | 30.56 | 31.67 |
| | M-SMoE | ✓ | 128 | 6.67 | 2.00 | 4.67 | 4.19 | 32.32 | 72.00 | 19.10 | 6.25 | 18.40 |
| | EASY-EP | ✓ | 128 | **75.94** | **61.98** | **42.50** | **57.63** | **70.36** | **91.20** | 57.95 | **34.17** | **61.47** |
| | Full | - | 256 | 55.73 | 47.71 | 28.75 | 48.50 | 66.87 | 87.51 | 64.22 | 33.33 | 54.08 |
| | Random | × | 64 | 0.00 | 0.00 | 0.00 | 0.00 | 26.87 | 0.39 | 0.00 | 0.69 | 3.49 |
| | Frequency | × | 64 | 31.35 | 34.06 | 15.73 | 1.95 | 45.25 | 40.13 | 61.96 | 22.74 | 31.65 |
| | Gating Score | × | 64 | 43.96 | 25.10 | 23.12 | 14.97 | 51.52 | 78.68 | 64.20 | 0.00 | 37.69 |
| | M-SMoE | × | 64 | 16.67 | 13.33 | 3.33 | 1.20 | 22.22 | 12.18 | 47.00 | 21.52 | 17.18 |
| | EASY-EP | × | 64 | **53.12** | **41.56** | **28.85** | **27.99** | **57.35** | **84.57** | **72.50** | **27.55** | **49.19** |
| DeepSeek | Random | × | 128 | 1.33 | 0.67 | 0.00 | 11.38 | 34.95 | 53.5 | 53.66 | 18.75 | 21.78 |
| -V3-0324 | Frequency | × | 128 | **55.73** | 42.60 | 30.10 | 36.08 | 63.54 | 84.29 | 66.84 | 31.71 | 51.36 |
| | Gating Score | × | 128 | 55.42 | 45.10 | 30.94 | **47.60** | 63.78 | 84.62 | **67.76** | 35.42 | 53.83 |
| | M-SMoE | × | 128 | 48.00 | 38.67 | 28.67 | 30.53 | 55.82 | 86.72 | 66.60 | 33.33 | 48.54 |
| | EASY-EP | × | 128 | 55.21 | **46.88** | **31.56** | 46.71 | **65.25** | 86.72 | 63.58 | **37.08** | **54.12** |
| | Frequency | ✓ | 128 | 51.35 | 37.60 | 24.27 | 17.07 | 55.90 | 83.47 | 66.80 | 36.25 | 46.59 |
| | Gating Score | ✓ | 128 | 53.75 | 40.10 | 27.19 | 28.74 | 58.88 | 83.86 | 67.74 | 34.58 | 49.36 |
| | M-SMoE | ✓ | 128 | 43.33 | 30.00 | 20.00 | 7.19 | 52.53 | 82.33 | 62.20 | 29.17 | 40.84 |
| | EASY-EP | ✓ | 128 | **57.81** | **46.56** | **33.33** | **40.72** | **64.95** | **85.00** | **72.26** | **38.74** | **54.92** |

**Baselines.** In our experiments, we employ three expert pruning methods and one expert merging method for comparison. For expert pruning, we employ different methods to assess the expert scores, including *random*, *frequency*, and *gating scores* (as discussed in Equation 2), and only keep Top-$M$ experts with the highest expert scores [4]. For expert merging, we select M-SMoE [25], which first employs neuron permutation alignment to mitigate neuron misalignment of experts and then merges experts into dominant ones with similarity of router logits.

## 5.2 Main Results

Table 2 showcases our method's performance against baselines under various pruning configurations. First, our approach consistently outperforms all baseline methods across diverse benchmarks and pruning settings. Notably, in domain-specific pruning, our method matches and even surpasses full model performance on certain benchmarks with only half the experts. This may be attributed to the effective removal of irrelevant experts, enhancing the model's ability to utilize domain-specific knowledge. Furthermore, our method demonstrates strong resilience to high compression ratios (*e.g.,* 75%), where most methods experience significant performance degradation, particularly on DeepSeek-R1. In contrast, our technique preserves substantial model capabilities, highlighting its effectiveness in identifying critical experts for specific tasks.

Second, non-reasoning models exhibit greater robustness compared to their reasoning-oriented counterparts. DeepSeek-V3-0324, for instance, retained more performance across most pruning methods than DeepSeek-R1. Under domain-specific pruning, DeepSeek-V3-0324's performance on datasets like AgentBench-OS even improved notably after pruning some experts. However, this phenomenon is not observed with DeepSeek-R1. We hypothesize that while domain capabilities might be preserved in pruned reasoning models, their long-term generation abilities are compromised.

---

[4]We do not include perturbation-based pruning methods in our comparison [6, 7], which is computationally prohibitive for MoE models with 256 experts (the detailed analysis is shown in Appendix E).

Finally, our method also excels in preserving performance under mixed-domain pruning. It retains over 90% of the original performance, surpassing domain-specific compression with other methods. Conversely, other expert pruning techniques struggle to maintain balanced performance across different domains. This underscores that the overlapped experts identified by our approach, which are linked to general reasoning abilities, effectively contribute to a broad array of downstream domains.

## 5.3 Detailed Analysis

In this section, we conduct further ablation studies and detailed analyses to investigate the effectiveness of our approach and the few-shot expert localization phenomena of large MoE models. For more analysis experiments, we present them in Appendix F.

### 5.3.1 Ablation Study

We conduct ablation studies to analyze the impact of each component in our method. Specifically, we evaluate two variants with 64 remaining experts: (1) removing the token-level contribution estimation and (2) replacing the product of gating values and L2 norm of expert outputs with only the gating scores. As shown in Table 3, both incorporating token-level contribution estimation and considering the L2 norm of expert outputs lead to improved performance compared to using only the gating score. Furthermore, combining both components results in the best overall performance. These findings highlight the importance of both components in our method.

Table 3: Results of ablation study. norm denotes whether considering L2 norm of expert output and Token denotes whether considering token contribution scores.

| Method | Metric | Experts | AIME-24 | AIME-25 | HMMT | LiveCode | GPQA | A-OS |
|---|---|---|---|---|---|---|---|---|
| Ours | $g_{i,t}^l \|e_{i,t}^l\| \cdot s_t^l$ | 64 | 72.81 | 55.33 | 36.00 | 42.51 | 67.47 | 27.26 |
| w/o Token | $g_{i,t}^l \|e_{i,t}^l\|$ | 64 | 65.33 | 49.33 | 31.33 | 27.54 | 56.57 | 21.53 |
| w/o norm | $g_{i,t}^l \cdot s_t^l$ | 64 | 70.00 | 40.00 | 23.33 | 19.76 | 61.11 | 18.75 |
| w/o both | $g_{i,t}^l$ | 64 | 2.67 | 1.33 | 2.67 | 0.00 | 20.20 | 0.69 |

### 5.3.2 Generalization Capacity

Beyond same-domain task evaluations, we assessed the generalization capacities on unrelated domains after domain-specific pruning (e.g., pruning DeepSeek-R1 with AIME2023 and evaluating on LiveCodeBench). As Table 4 shows, the model demonstrates a certain generalization ability, especially in similar domains (*e.g.,* math/science, code/OS-agent, science/medicine). We hypothesize that while some domain-specific experts are pruned, core reasoning experts are preserved. However, the out-of-domain performances are typically poorer than mixed-domain pruning. Thus, we suggest using mixed-domain pruning when facing multiple downstream tasks.

Table 4: Results of generalization capacities of pruned models. Domain denotes the domain of pruning data. **Bold** denotes in-domain performances.

| Domain | AIME24 | LiveCodeBench | GPQA | Agent-OS | USMLE | FinIQ |
|---|---|---|---|---|---|---|
| Math | **79.17** | 46.11 | 46.91 | 3.47 | 46.43 | 58.20 |
| Coding | 38.00 | **61.11** | 39.90 | 15.97 | 41.79 | 53.00 |
| Science | 64.64 | 53.59 | **70.12** | 4.17 | 75.88 | 57.50 |

### 5.3.3 Effect of Pruning Data

We also study the impact of pruning data. Instead of using the full model's input and output, we examine five types of data: (1) *Inp*: just input context data; (2) *Out*: only the model's generated data; (3) *Inp+Ans*: input context data and the correct generated answer; (4) *PT*: pre-training data from the same domain; and (5) *CC*: data from CommonCrawl. Table 5 and 6 show the experimental results for DeekSeek-R1 and DeepSeek-V3-0324. Compared to using the combination of input and model output, performance decreased under other settings for both math and coding tasks. This suggests that,

even within the same domain, there are notable differences in expert distributions among data types. Additionally, employing CommonCrawl data causes significant performance declines, demonstrating that pruning a task-irrelevant model with pre-training data is not suitable for current models.

Table 5: Comparison of performances of DeepSeek-R1 with different data.

| Data | AIME24 | LiveCodebench | GPQA |
|---|---|---|---|
| Inp+Out | 79.17 | 61.11 | 70.12 |
| Inp+Ans | 75.33 | 53.89 | 67.98 |
| Inp | 66.00 | 50.90 | 70.81 |
| Out | 77.33 | 59.28 | 70.10 |
| PT | 29.33 | 38.32 | 66.16 |
| CC | 0.00 | 0.00 | 28.92 |

Table 6: Comparison of performances of DeepSeek-V3-0324 with different data.

| Data | AIME24 | LiveCodebench | GPQA |
|---|---|---|---|
| Inp+Out | 55.73 | 48.50 | 66.87 |
| Inp+Ans | 54.00 | 43.11 | 61.62 |
| Inp | 51.33 | 14.97 | 63.64 |
| Out | 54.67 | 47.90 | 63.13 |
| PT | 13.33 | 37.72 | 56.57 |
| CC | 0.00 | 0.00 | 29.80 |

#### 5.3.4  Effect of Number of Pruning Demonstrations

In Section 3.2, we observe that domain-relevant experts can be identified with only a few demonstrations. Here, we further investigate the impact of the number of demonstrations on final performance. To do so, we sample varying numbers of demonstrations from the same distribution and prune half of the experts in each layer. The performance variations on AIME24 and LiveCodeBench are presented in Figure 4. When we utilize only a single sample for pruning, the selected experts are often influenced by the characteristics of that individual sample, thus resulting in lower performance. As we further increase the number of demonstrations, the performance rapidly rises, achieving comparable performances with the full model.

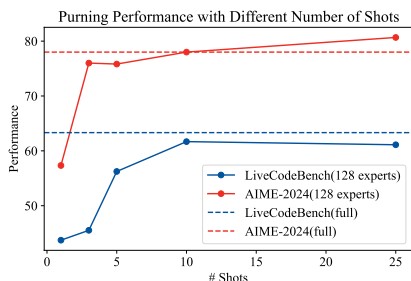

Figure 4: Comparison of performance with different numbers of shots for pruning.

#### 5.3.5  Analysis of Throughput

To evaluate the throughput of pruned models with different numbers of experts, we use the SGLang [26] package and measure performance under a maximum request concurrency of 32. We evaluate the settings with 1K input and 1K output length. For configurations with more than 192 experts, two 8×H800 GPUs are used. Figure 1 shows the scaling throughput for this setting. We observe that reducing the number of experts significantly improves throughput, particularly when the model can be deployed on a single node. Compared to the full DeepSeek-R1 model, configurations with 128 and 64 experts achieve 2.99× and 4.33× throughput, respectively. Compared to the full model, the pruned model can be deployed on a single node, thereby avoiding inter-node communication overhead. Moreover, using fewer experts further reduces communication between GPUs within the node, improving computational efficiency. We provide more experiments in Appendix F.5.

## 6  Related Work

MoE architectures improve computational efficiency by activating only a small subset of experts per input [27, 28, 4]. However, the increased number of parameters introduced by these architectures leads to substantial memory overhead. To address this, existing efforts broadly fall into two categories. The first line of work focuses on architectural optimization to reduce computation or parameter size. Representative methods include pyramid-shaped expert allocation [29], fine-grained expert design with smaller per-expert modules [14], and the transformation of dense models into sparse MoE variants [30–32]. While effective, these approaches typically require modifying model architecture and retraining from scratch, which limits their applicability to already deployed or pretrained models. The second line of research approaches the problem from a memory efficiency perspective by applying post-hoc compression techniques, primarily pruning and quantization [33–35]. These

methods typically estimate expert importance based on routing frequency [5], gating scores [36], or direct measurement of their contribution to model outputs [6, 7]. Some approaches further reduce redundancy by merging similar experts [8, 9], though they may meet the problem of neuron misalignment and additional memory costs [10, 11]. In contrast, our method enables efficient pruning with a single forward pass and avoids storing additional model variants during compression.

# 7  Conclusion

In this work, we investigated the domain specialization of experts in large MoE models. Our observations indicate that domain-specific experts play a crucial role in their respective domains and can be effectively identified with a few demonstrations. Building on these insights, we proposed a pruning strategy that leverages demonstrations from tasks within the same domain. Specifically, we introduced EASY-EP, a simple yet effective pruning method that combines output-aware expert importance assessment with expert-level token contribution estimation. Experimental results showed that our approach maintained comparable performance while utilizing only half of the experts in domain-specific settings and retained over $90\%$ of the original performance in mixed-domain pruning. We believe that our method can facilitate the deployment of large MoE models, particularly for efficiently handling a high volume of samples within the same domain.

# 8  Limitation

Our work investigates the phenomenon of few-shot expert localization in large MoE models and proposes a simple yet effective method for domain-specific expert pruning. We leave the investigation of training to further enhance the pruned model's performance, particularly in balancing in-domain and out-of-domain capabilities, as future work, given our current focus on evaluating pruning effectiveness under realistic resource constraints. We observed differing levels of robustness to expert pruning between reasoning-oriented (DeepSeek-R1) and non-reasoning models (DeepSeek-V3-0324). While this trend is noteworthy, further validation on a broader range of architectures is needed to strengthen the generality of this observation.

# Acknowledgments

This work was partially supported by National Natural Science Foundation of China under Grant No. 92470205 and 62506077, Beijing Natural Science Foundation under Grant No. L233008 and Beijing Municipal Science and Technology Project under Grant No. Z231100010323009. Peiyu Liu and Xin Zhao are the corresponding authors.

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

# A  Expert Overlap Across Domains

To further analyze the overlap of top experts across different domains, we select AIME-2023, LiveCodeBench-V3, and GPQA as calibration sets and select top experts with gating scores. Subsequently, we compute the overlap ratio of Top-$M$ experts ($M \in \{1, 2, 4, 8, 16, 32, 64, 128\}$) and expert overlap across different layers, which are shown in Figure 5. We can observe that the experts with the highest gating scores differ significantly across different domains, and as the number of top experts increases, the overlap ratio increases. Additionally, in the lower layers, there is a higher overlap among experts with significant gating scores across different datasets, but this overlap becomes relatively lower in the middle and deeper layers. This indicates that experts in the lower layers tend to focus on general capacities, while as the network depth increases, they gradually specialize in handling knowledge from distinct domains.

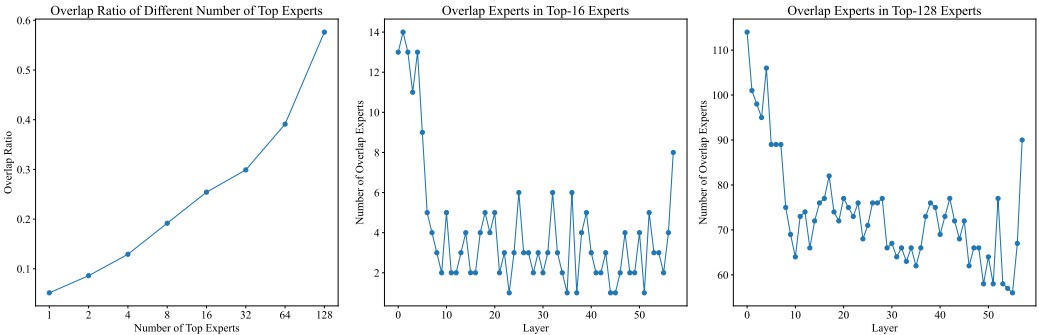

Figure 5: Left: Overlap ratios of Top-$M$ experts with different $M$. Middle and Right: Overlap experts in top-16 and 128 experts on different layers.

# B  Empirical Study with EASY-EP

To further illustrate the phenomenon of few-shot expert localization, we analyze it with the expert scores (as discussed in Equation 3) from our proposed EASY-EP method in place of the gating scores.

## B.1  Expert Specialization Across Domains

**Distinct Expert Distribution Across Domains.** We compute the overlap ratios of the top experts identified by our method and visualize them in Figure 6. Similar to previous observations, large differences in expert distribution still exist across different domains. However, the expert overlap between domains is larger than that of experts identified by gating scores. This may be because our method identifies experts not only frequently activated within a domain but also those who contribute more than other experts in residual connections. The latter may be more similar across different datasets, thus leading to higher overlap.

**Impact of Removing Domain-Specific Experts.** Similarly, we remove experts whose score, as calculated by our method, falls within the top-128 for a single domain but not for any other. As shown in Table 7, removing these in-domain experts leads to larger performance degradation than out-of-domain experts. However, compared with pruning with gating scores, the performance degradation is less severe. We speculate this is due to the higher overlap among the top experts identified by our method, resulting in a smaller pruning ratio.

## B.2  Expert Locality Within One Domain

**Effect of Calibration Set Size.** We present the overlap ratios of top-128 experts selected via our method with different numbers of demonstrations. As shown in Figure 7 (Left), as the number of demonstrations increases, the identified domain-specific experts gradually become stable. For 25-shot

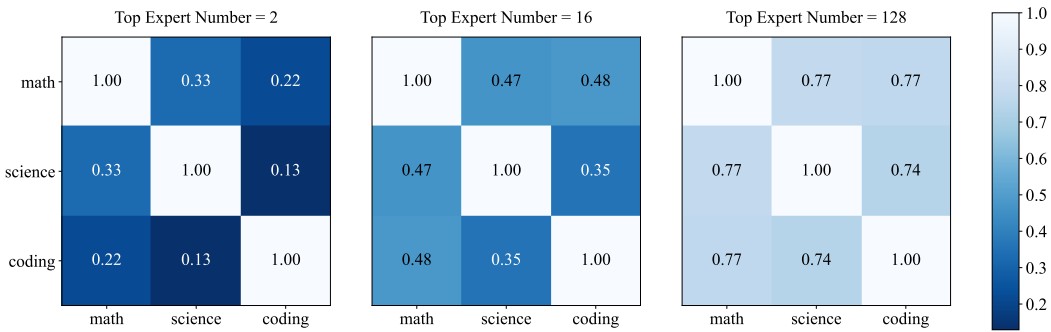

Figure 6: Overlap ratios of experts with top expert scores computed with our methods on different datasets.

Table 7: Results of removing domain-specific experts with EASY-EP.

| Domain | AIME24 | GPQA | LiveCodeBench |
|--------|--------|------|---------------|
| Full | 77.08 | 70.91 | 63.32 |
| Math | 72.67 (**-3.41**) | 73.17 (+2.26) | 64.22 (+0.90) |
| Code | 76.67 (-0.41) | 72.93 (+2.02) | 62.27 (**-1.05**) |
| Science | 76.00 (-1.08) | 58.23 (**-12.68**) | 64.97 (+1.65) |

pruning, it can achieve the overlap ratio of $93\%$ with 10-shot pruning, further demonstrating the feasibility of few-shot domain-specific expert pruning.

**Consistency of Expert Activation Across Datasets.** Figure 7 (Right) illustrates that the top-128 experts identified by our method exhibit consistent overlap ratios across different datasets within the math domain. Our method not only presents expert consistency over different datasets similar to the gating score metric, but also yields significantly higher overlap ratios. We hypothesize that our approach is more adept at identifying experts of true domain-wide significance, as opposed to those who are only important in a single dataset due to the dataset-specific differences.

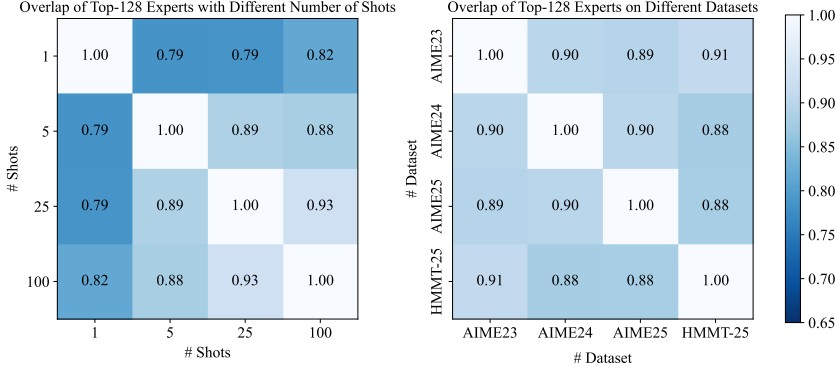

Figure 7: Left: Overlap ratio of top-128 experts identified by EASY-EP with different numbers of demonstrations. Right: Overlap ratio of top-128 experts identified by EASY-EP pruned with different math datasets.

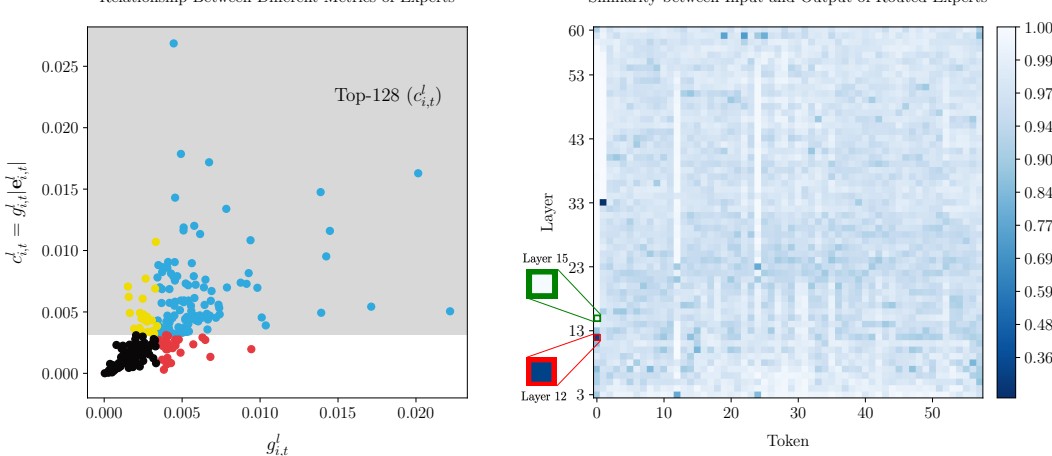

Figure 8: Left: Gating scores and averaged product of gating value and L2 Norm of expert outputs. Blue/red dots indicate experts in top-128 gating scores $g_{i,t}^l$; blue/yellow dots denote experts in top-128 expert importance $c_{i,t}^l$; black dots indicate neither. Right: Consine similarity between representations before and after incorporating the outputs of the routed expert. The red box and green box indicate high similarity and low similarity, respectively.

## C  Empirical Analysis of Components in EASY-EP

### C.1  Output-Aware Expert Importance Assessment

To analyze the relationship between whether a large gating score $g_{i,t}^l$ ensures a large product of gating scores and L2 norm of expert output $g_{i,t}^l \|e_{i,t}^l\|$, we visualize the relationship of top-128 experts selected by the two metrics. As shown in Figure 8 (Left), despite a significant degree of overlap among experts, there still exist some experts who excel exclusively in a single, focused metric. Although the gating scores of some experts are not large, the L2 norms of their outputs are larger than others. This proves the necessity of considering experts' outputs.

### C.2  Expert-Level Token Contribution Estimation

Given one sample selected from AIME-2023, we first obtain representations before and after incorporating the outputs of routed experts in each layer of DeepSeek-R1 and compute the similarities between them, as illustrated in Figure 8 (Right). We can observe that the similarities differ significantly across tokens and layers. For some tokens at specific layers, skipping the expert module results in minimal changes to the hidden states, with over $99\%$ similarity between the input and output representations (*e.g.,* the 15th layer of the first tokens). In contrast, for certain tokens and layers (*e.g.,* the 12th layer of the first tokens), the hidden states show substantial differences after expert routing.

## D  Experiment Details

As shown in the previous analysis in Section 3.2, datasets within the same domain can identify the important experts on other datasets. Thus, for different domains and tasks, we select one dataset as a calibration set, which is shown in Table 8. For mixed-domain pruning, we choose the scores calculated on each 25-shot calibration set and average the normalized scores as the final scores of experts. For the evaluation of FinanceIQ, we randomly select 1000 samples with the seed of 42 since the test set is too large.

Additionally, all the experiments are conducted in one $8\times$ H200 GPU. We set the maximum context length to 32K, the temperature to 0.6, and the top-p sampling value to 0.95 for most benchmarks (temperature as 0.2 for LiveCodeBench). To ensure statistical reliability, most benchmark is evaluated independently 5 times (32 times for math benchmarks), and we report the average performance of pass@1.

Table 8: Calibration Set of Each Domain

| Domain | Calibration Set | License |
|--------|----------------|---------|
| Math | AIME 2023 | cc-by-nc-sa-4.0 |
| Coding | LiveCodeBench-V3 | MIT |
| Science | GPQA-Main | MIT |
| Agent | Dev Set of AgentBench-OS | Apache-2.0 license |
| Finance | Dev Set of FinanceIQ | MIT |
| Medical | Dev Set of USMLE | cc-by-nc-sa-4.0 |

# E  Analysis of Perturbation-based Pruning Method

Previous studies [6, 7] have proposed methods that utilize representation perturbation after expert pruning to determine which experts should be removed. In NAEE [6], identifying the optimal subset of experts requires $C_N^{N'}$ evaluations, where $N$ and $N'$ represent the original and target numbers of experts, respectively. In CD-MoE [7], a greedy search algorithm selects the expert to retain based on minimal representation perturbation through a rolling mechanism, requiring $N(N + N')/2$ evaluations. For DeepSeek-R1, which has 256 experts and a target expert count of 128, these methods require over $10^{75}$ and $24768$ evaluations per layer, respectively. Thus, the perturbation-based methods are not affordable for MoE models with a large number of experts. Conversely, our method only requires one forward operation to identify critical experts, which is cheaper and more suitable for large MoE models.

# F  Further Experimental Analysis

## F.1  Experiments with Qwen3-30B-A3B

To further demonstrate the effectiveness of our method, we evaluate the performance of Qwen3-30B-A3B with different expert pruning methods. Specifically, the expert number of the model is pruned from 128 to 64. As shown in Table 9, our method can achieve significantly better performances than other methods, which demonstrates the effectiveness of our method. Additionally, compared with the DeepSeek series, it is harder to compress Qwen3-30B-A3B since it has no shared experts.

| Qwen-30B-A3B | AIME24 | AIEM25 | HMMT | GPQA | LiveCodeBench | Averge |
|--------------|--------|--------|------|------|---------------|--------|
| FULL | 80.42 | 70.83 | 50.00 | 68.59 | 62.80 | 66.53 |
| Random | 0.00 | 0.00 | 0.00 | 23.23 | 0.00 | 4.65 |
| frequency | 13.33 | 10.00 | 0.00 | 61.11 | 14.00 | 19.69 |
| Gating Scores | 13.33 | 6.67 | 6.67 | 35.35 | 14.00 | 15.20 |
| EASY-EP | 72.92 | 61.25 | 38.75 | 63.13 | 46.30 | 56.47 |

Table 9: Experiments on Qwen3-30B-A3B.

## F.2  Expert Overlap with Different Metrics

We calculate the pairwise overlap of the top-128 experts chosen by various metrics across three datasets: AIME23, GPQA-main, and LiveCodeBench-V3. The results are presented as a heatmap in Figure 9. Our analysis reveals that the experts selected based on gating scores and frequency exhibit a remarkably high overlap, approximately $90\%$. However, the experts identified by our method show significant divergence from these metrics, underscoring the influence of incorporating expert outputs and residual changes in our approach. We posit that the large discrepancy among experts results in a huge performance gap between models.

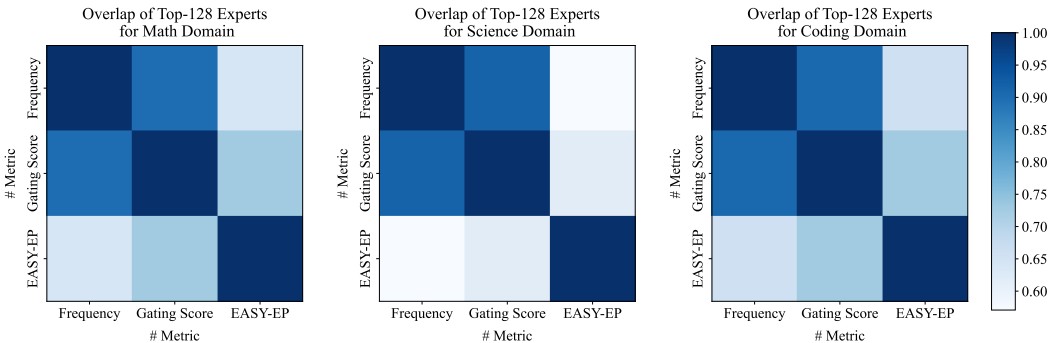

Figure 9: Overlap of Top-128 Experts Selected via Different Metrics.

| Model | AIME24 | AIME25 |
|---|---|---|
| FULL | 80.42 | 70.83 |
| EASYEP | 72.92 | 61.25 |
| EASYEP w.o. reroute | 73.33 | 63.33 |

Table 10: Performances of removing experts without rerouting.

## F.3 Effect of Directly Removing Experts Without Reroute

In addition to our settings, where we remain the same number of activated experts, we also experimented with computing the gating weights for each expert as normal and selecting them accordingly. Then, for the pruned experts, we set their outputs to zero. Thus, to some extent, tokens that would have been routed to the pruned experts are processed by fewer experts. We employ Qwen3-30B-A3B for experiments and compare the performances with and without reroute, as shown in Table 10. We can observe that the performances slightly improve compared to the original version. In addition, it can save computation due to the reduced number of activated experts, which can further accelerate the inference speed.

## F.4 Results of Pruning with Different Ratios at Different Layers

To further investigate the pruning performance of our method, we employed layer-wise dynamic pruning. Specifically, we first normalized the expert scores for each layer and ranked all experts across all layers. Subsequently, we pruned the top experts using different pruning ratios. As shown in Table 11, we observe that employing layer-wise dynamic pruning does not always lead to better performance, and the performance on LiveCodeBench even drops significantly. In addition, the dynamic compression ratio leads to deployment difficulties, as it requires different numbers of experts for each layer. Thus, we suggest just employing a fixed pruning ratio across all layers.

Table 11: Performance comparison of EASY-EP with and without employing layer-wise compression ratios.

| Model | Ratio | Layer | AIME2024 | GPQA | LiveCodeBench |
|---|---|---|---|---|---|
| DeepSeek-R1 | 0.5 | × | 79.17 | 70.12 | 61.11 |
| | 0.5 | ✓ | 79.33 | 65.15 | 44.31 |
| | 0.25 | × | 72.67 | 67.47 | 42.51 |
| | 0.25 | ✓ | 65.33 | 66.16 | 28.74 |
| DeepSeek-V3-0324 | 0.5 | × | 55.21 | 65.25 | 46.71 |
| | 0.5 | ✓ | 58.67 | 62.63 | 38.23 |
| | 0.25 | × | 53.12 | 57.35 | 27.99 |
| | 0.25 | ✓ | 58.67 | 60.10 | 21.56 |

## F.5 Throughput with Different Lengths

To further explore the throughput changes on pruning models, we consider three additional length settings: (1) *1K+4K*: 1K input length and 4K output length; (2) *4K+1K*: 4K input length and 1K output length; and (3) *4K+4K*: 4K input length and 4K output length. Figure 10 (Right) presents results for the other settings. Compared with *1K+1K*, either increasing the length of input or output leads to lower throughput and throughput acceleration after pruning. Additionally, in long output scenarios with an equivalent total sequence length, the 128-expert configuration achieves a significantly higher acceleration ratio (2.91× under the *1K+4K* setting, compared to 2.52× under the *4K+1K* setting).

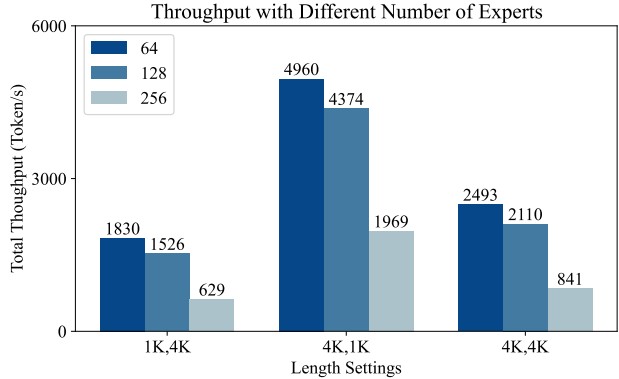

Figure 10: Total throughput across different numbers of experts.

