# OpenReview forum: "Domain-Specific Pruning of Large Mixture-of-Experts Models with Few-shot Demonstrations"
_NeurIPS.cc/2025/Conference — NeurIPS 2025 poster_

### Official Review · Reviewer_3TmE · 2025-06-16

**Clarity:** 3
**Significance:** 2
**Originality:** 2
**Rating:** 3
**Confidence:** 4

**Summary:**

This paper proposes EASY-EP, a training-free pruning framework that (i) scores each expert by the product of its gating value and output norm and (ii) weights scores by token-level representation change, then retains only the top-ranked experts per layer. Applied to DeepSeek-R1 and DeepSeek-V3-0324, pruning 50 % of experts maintains ≥90 % of full-model accuracy across math, coding, science, finance, medicine, and agent tasks, while boosting 1k-token throughput by 2.99× and allowing single-node deployment. EASY-EP needs just one forward pass, no retraining, and also supports mixed-domain pruning with balanced performance, enabling memory- and latency-efficient use of gigantic MoE models.

**Questions:**

1. The paper acknowledges a larger accuracy drop when pruning with a single domain and testing on a different one. Could you add qualitative or quantitative analyses (e.g., expert-activation overlap heat-maps) to pinpoint why EASY-EP underperforms here and outline mitigation strategies such as hybrid expert pools or dynamic gating?

2. In line 121, could the authors properly define "demonstrations"?

**Ethical Concerns:**

["NO or VERY MINOR ethics concerns only"]

**Limitations:**

No discussion on limitations. The authors could consider discussing whether the insights of this work can be extended to much boarder domains/modalities/MoE architectures.

**Quality:**

3

**Strengths And Weaknesses:**

**Strengths:**

1. The empirical study spans two large-scale SOTA LLMs with MoE architecture and six downstream domains. The comprehensive ablations verify the necessity of both scoring factors that the paper proposed.

2. The method is straightforward: one forward pass suffices, avoiding the combinatorial evaluations required by perturbation-based baselines.

3. This paper provides a lot of helpful insights, such as MoE routers naturally localize to a small, domain-consistent expert set with only a handful of in-domain examples. It further shows that a single-node deployment with minimal accuracy loss is practically valuable.

**Weakness:**

1.	The proposed method is largely incremental and offers limited structural novelty; its computations rely primarily on similarity or norm-based measures.

2.	To enhance interpretability, the authors could analyze expert capacities and token-routing distributions across different domains, comparing these patterns between the baseline and the proposed approach.

3.	The study’s statistical reliability is weak: error bars and significance tests are absent, leaving uncertainty about the robustness of the reported improvements.

4. It would be better to validate the performance of the method on at least one additional MoE family other than DeepSeek, e.g. Qwen or Mixtral. This would be helpful to show that the method can generalize beyond DeepSeek’s router design

---

> ### Author Rebuttal · Authors · 2025-07-31
>
> Thank you for your insightful suggestions!
>
> ## W1: limited structural novelty
>
> We appreciate the reviewer’s concern about the structural novelty of our approach and the reliance on similarity- or norm-based measures.
>
> (1) First, our goal is not to introduce architectural changes but to provide a practical pruning framework tailored to ultra-large MoE models. In this context, structural changes in the sense of redesigning the MoE architecture would require retraining from scratch, which is infeasible at the scale of DeepSeek-R1 (671B). Instead, we focus on a post-hoc pruning strategy that is lightweight, training-free, and easily deployable, which we argue is a critical form of innovation for models of this magnitude.
>
> (2) Second, selecting an effective metric for expert pruning is non-trivial， especially for large MoE models. Through our analysis on DeepSeek-R1, we observed a counterintuitive phenomenon: with ultra-large MoEs, a small number of calibration samples is sufficient to consistently identify domain-specialized experts. We term this the *few-shot expert localization* phenomenon. Building on this, our integration of output-aware expert importance (gating value × output norm) with token-level contribution estimation enables us to capture both the strength and the influence of expert activations in a single forward pass, making accurate and efficient pruning feasible at this unprecedented scale.
>
> (3) Third, our scoring strategy is theoretically grounded. Beyond empirical observations, we show that using the L2 norm of the gated expert output and similarity of the input and output of MoE modules can be interpreted as an approximation of the gradient-based contribution to the loss. This ensures that our pruning strategy is not heuristic but aligned with the underlying optimization objective. We show the analysis below.
> ### Optimization Objective:
>
> Our pruning goal is to minimize the expected loss on a calibration dataset $D_{\text{calib}}$, under a constraint that only $M$ experts are retained per layer. Formally, we solve:
>
> $$
> \min_{m_l} \mathbb{E}^{(x,y)}  (\mathcal{L}(f(x; \{m_l\}), y)) , \quad \text{s.t. } \sum_{i=1}^N m_{il} = M
> $$
>
> Here, $m_{il} \in \{0, 1\}$ is a binary mask indicating whether expert $i$ in layer $l$ is retained.
>
> This is a combinatorial problem and is NP-hard. To solve it approximately, we adopt a greedy strategy that relies on estimating the importance of each expert.
>
> ### Gradient-Based Expert Importance:
>
> We approximate the change in loss when removing expert $j$ via a first-order Taylor expansion:
>
> $$
> \Delta \mathcal{L} \approx -\frac{\partial \mathcal{L}}{\partial m_{jl}}
> $$
>
> Applying the chain rule, we obtain:
>
> $$ \frac{\partial \mathcal{L}}{\partial m_{j,l}} = \sum_{t=1}^T \left( \frac{\partial \mathcal{L}}{\partial \tilde{h}{t,l}} \right)^T \left( g_{j,t,l} e_{j,t,l} \right) $$
>
> This shows that the importance of expert $j$ is proportional to the projection of the gradient of the loss onto the expert’s output.
>
> ### L2 Norm in Output-Aware Expert Importance Assessment:
>
> While we cannot compute the gradient $\partial \mathcal{L} / \partial \tilde{h}\_{t,l}$ during a forward-only pass, we **can** compute the expert’s contribution term $g\_{j,t,l} e\_{j,t,l}$.
>
> $$
> ( \frac{\partial \mathcal{L}}{\partial \tilde{h}\_{t,l}} )^\top  g\_{j,t,l} e\_{j,t,l} ) \leq \| \frac{\partial \mathcal{L}}{\partial \tilde{h}\_{t,l}} \|\_2 \cdot \| g\_{j,t,l} e\_{j,t,l} \|\_2
> $$
>
> This upper bound shows that **the L2 norm of $g_{j,t,l} e_{j,t,l}$** is a direct proxy for the expert’s contribution to the loss-sensitive direction in the model. Since the gradient component is unavailable during a forward pass, we approximate expert importance using this measurable term.
>
> Thus, **using the expert’s output L2 norm as part of the importance score is not ad hoc—it follows directly from a first-order gradient-based analysis of our objective.**
>
> ### Additional analysis of Expert-Level Token Contribution Estimation
> For expert-token contribution estimation, we need to compute $\partial \mathcal{L} / \partial \tilde{h}_{t,l}$. However, it is very costly to conduct gradient computation. Thus, we emply $1-Sim(\tilde h_t^l,h_t^l)$ as a approximation.
>
>
> ## W2: Expert capacities and token-routing distributions across different domains with different methods.
>
> In Figures 2, 6, and 9, we present a comparison of the similarities between the gating scores and EASYEP experts across different domains, as well as the similarities among EASYEP, gating scores, and frequency. It can be observed that our method exhibits greater variability in expert importance compared to the others, allowing it to better capture the relative importance among experts. On the other hand, our method shows higher similarity across domains than the gating scores, which to some extent indicates that our approach captures expert importance more effectively, rather than merely domain-specific characteristics, thereby maintaining a stronger overall capability.
>
> ## W3: Weak statistical reliability
>
> We have provided the mean and standard deviation for computing error bars of DeepSeek-R1 on the AIME2024 dataset (32 samples), as shown in the table below. The results demonstrate that our method achieves better performances and comparable robustness  compared to the full model baseline.
>
> | **Model** | **Mean Accuracy** | **Standard Deviation** |
> | --- | --- | --- |
> | Full Model | 0.7708 | 0.0232 |
> | Our Method | 0.7917 | 0.0289 |
>
> ## W4: Experiments with more models
>
> We provide the experiments results on Qwen3-30B-A3B, the results are shown in the following Table. We can observe that for the MoE models without shared experts (which appears in DeepSeekMoEs), expert pruning is much more hard. For example, performances of other methods on AIME is very poor. On the contory, our method can achieve over 80% performances on AIME with only 50% experts, further demonstrating the effectiveness and universality.
>
> | Qwen-30B-A3B | AIME24 | AIEM25 | HMMT | GPQA | LiveCodeBench |
> | --- | --- | --- | --- | --- | --- |
> | FULL | 80.42 | 70.83 | 50 | 68.59 | 62.8 |
> | Random | 0 | 0 | 0 | 23.23 | 0 |
> | frequency | 13.33 | 10 | 0 | 61.11 | 14 |
> | Gating Scores | 13.33 | 6.67 | 6.67 | 35.35 | 14 |
> | EASYEP | 72.92 | 61.25 | 38.75 | 63.13 | 46.3 |
>
> ## Q1: **Analysis of performance drops of task-specific pruning and methods to mitigate it.**
>
> We agree that pruning with a single-domain focus can lead to performance degradation when testing on different domains, and we have performed both qualitative and quantitative analyses to better understand and address this limitation of EASY-EP.
>
> **Low Cross-Domain Expert Overlap**: Figure 2 (A) and (B) in our paper provide direct evidence. When examining the top 128 experts ranked by gating scores, the overlap between the Math and Code domains is only 74%. For the top 16 experts, the overlap is almost negligible, with nearly disjoint sets.
>
> **Essential Role of Domain Experts**: Removing Science-specific experts causes GPQA accuracy to drop by **11.82 points**. In contrast, the same pruning leads to only a **2.25-point** decrease on LiveCodeBench, a Code task. This asymmetry highlights that domain experts are critical within their respective domains but have limited transferability.
>
> Ways to mitigate it:
>
> ● **Mixed-Domain Pruning:** By pruning with mixed-domain data, we can preserve a broader range of important experts. As shown in our paper, it can achieve a great performance preservation on multiple tasks.
>
> ● **Task-Aware Expert Activation**: using a lightweight classifier or predictor to identify the domain or task of the input, and then selectively activating the corresponding experts.
>
> ## Q2: In line 121, could the authors properly define "demonstrations"?
> ``Demonstrations'' is a common term in In-Context Learning (ICL). Specifically, it refers to examples or sample inputs that are provided to an LLM. These samples serve as references that guide the model's responses or predictions.

---

> > ### Comment · Reviewer_3TmE · 2025-08-06
> > **Reply to Rebuttal**
> >
> > I greatly appreciate the authors' response, which has addressed a large portion of my concerns. However, after careful consideration and taking into account the views of other reviewers, I have decided to uphold my initial score.

---

> ### Author Response · Authors · 2025-08-05
> **A friendly reminder that our discussion period ends soon**
>
> Dear reviewer,
>
> This is a kind reminder as the discussion period will end soon. We have made every effort to thoroughly address your comments in our responses, including conducting additional experiments and providing further analysis. If you have any remaining questions or concerns, please let us know at your earliest convenience.

---

### Official Review · Reviewer_mHRa · 2025-07-02

**Clarity:** 2
**Significance:** 2
**Originality:** 2
**Rating:** 4
**Confidence:** 3

**Summary:**

The paper investigates the challenges associated with the memory overhead of large Mixture-of-Experts (MoE) models, particularly focusing on the DeepSeek-R1 model, which has 671 billion parameters. The authors introduce a novel pruning framework called EASY-EP, which leverages a few domain-specific demonstrations to effectively identify and retain the most relevant experts while discarding less important ones.

The key contributions of the paper include:

1. Observation of Few-Shot Expert Localization: The authors identify a consistent behavior where, given a small number of in-domain examples, the MoE model activates a stable and sparse subset of experts that are highly relevant to the task at hand. This phenomenon is termed "few-shot expert localization."

2. Development of EASY-EP Framework: The proposed framework consists of two main components: output-aware expert importance assessment and expert-level token contribution estimation. This allows for efficient pruning of experts based on their relevance to the current task, significantly reducing memory requirements without sacrificing performance.

3. Empirical Validation: The authors conduct extensive experiments on the DeepSeek-R1 and DeepSeek-V3-0324 models across various benchmark datasets. They demonstrate that their pruning method can maintain comparable performance while achieving a 2.99 times increase in throughput under the same memory constraints by retaining only half of the experts.

4. Domain Specialization Insights: The paper provides insights into the domain specialization of experts within large MoE models, showing that experts are often highly specialized and can be reliably identified with minimal demonstrations. This finding underscores the potential for domain-specific pruning strategies in large-scale models.

Overall, the paper presents a significant advancement in the efficient deployment of large MoE models, addressing both memory constraints and performance optimization through innovative pruning techniques.

**Questions:**

1. Statistical Significance Reporting: The paper does not provide error bars or any information regarding the statistical significance of the experimental results. This is a critical aspect of scientific reporting, as it helps to understand the reliability and variability of the results. The authors should include this information to strengthen their claims and provide a clearer picture of the model's performance.
2. Reproducibility of Results: While the paper claims to provide sufficient information for reproducing the results, it would benefit from a more detailed description of the experimental setup, including specific hyperparameters, training procedures, and the exact datasets used. This transparency is essential for other researchers to validate the findings and build upon the work.
3. Impact of Expert Pruning on Long-term Generation Abilities: The paper mentions that while domain capabilities may be preserved in pruned models, there are concerns about the long-term generation abilities being compromised. The authors should provide more empirical evidence or analysis to support this claim. It would be helpful to include experiments that specifically assess the long-term performance of the pruned models compared to the full models.
4. Generalization Across Domains: The concept of few-shot expert localization is intriguing, but the authors should elaborate on how well the identified experts generalize across different domains. It would be beneficial to include additional experiments that test the model's performance on unseen datasets within the same domain to validate the robustness of the few-shot approach.
5. Clarity and Structure of the Paper: The paper could improve its clarity by ensuring that the key contributions and findings are clearly articulated in the abstract and introduction. A more structured presentation of the results, including visual aids or tables summarizing key findings, would enhance the reader's understanding and engagement with the material.

These suggestions aim to enhance the rigor and clarity of the paper, ultimately contributing to its impact in the field.

**Ethical Concerns:**

["NO or VERY MINOR ethics concerns only"]

**Final Justification:**

The author addressed most of my concerns, however, after careful consideration and reference to the comments of other reviewers and the author's responses, such as whether the proposed combination of indicators was a suitable solution for the described scenario, etc., I decided to maintain my initial score

**Limitations:**

The paper presents a thorough investigation into the domain specialization of experts in large Mixture-of-Experts (MoE) models, particularly focusing on the DeepSeek-R1 model. However, there are several limitations that should be addressed:

1. Generalizability of Findings: While the study demonstrates the effectiveness of the proposed EASY-EP pruning method across various domains, it primarily relies on specific datasets (e.g., AIME-2023, GPQA-main, LiveCodeBench-V3). The authors should discuss how well these findings generalize to other domains or tasks not covered in their experiments.
2. Long-term Performance: The paper notes that while domain capabilities may be preserved in pruned models, there are concerns about the long-term generation abilities being compromised. This aspect requires further exploration, as it could significantly impact the practical deployment of the model in real-world applications.
3. Statistical Significance: The paper does not report error bars or other statistical significance measures for the experimental results. Including this information would strengthen the validity of the claims made regarding the performance of the proposed method.
4. Potential Negative Social Impacts: The authors state that there is no societal impact of the work performed. However, it is essential to consider potential negative implications, such as the misuse of the technology or biases that may arise from the model's training data. A more in-depth discussion on these aspects would be beneficial.
5. Clarity and Detail in Methodology: While the methodology is described, there could be more clarity regarding the specific steps taken during the pruning process and how the calibration sets were selected. This would enhance reproducibility and understanding of the method.

In summary, while the paper makes significant contributions to the field, addressing these limitations and suggestions would enhance its robustness and applicability.

**Paper Formatting Concerns:**

There is no incorrect formatting

**Quality:**

2

**Strengths And Weaknesses:**

Strengths:
1. The paper presents a well-structured analysis of expert activations in large Mixture of Experts (MoE) models, particularly focusing on the DeepSeek-R1 model. The methodology is robust, employing a novel pruning framework (EASY-EP) that effectively leverages few-shot demonstrations to identify and retain relevant experts. The experimental results demonstrate that the proposed method achieves comparable performance while significantly increasing throughput, indicating a high-quality contribution to the field.
2. The document is clearly organized, with a logical flow from the introduction to the methodology and results. Key concepts such as few-shot expert localization and the components of the EASY-EP framework are well-defined, making it accessible to readers with varying levels of expertise in the subject matter. The use of figures and tables to illustrate findings enhances understanding.
3. The research addresses a critical challenge in deploying large-scale MoE models—memory efficiency. By demonstrating that domain-specific experts can be effectively identified with minimal demonstrations, the paper contributes valuable insights that could lead to more efficient model deployment in real-world applications. The findings have implications for various domains, including math, science, and coding, which adds to the paper's significance.
4. The concept of few-shot expert localization is a novel contribution to the field of machine learning and MoE architectures. The integration of output-aware expert importance assessment and expert-level token contribution estimation into the pruning process showcases innovative thinking and a fresh approach to expert pruning strategies.

Weaknesses:

1.While the experimental results are promising, the paper lacks detailed statistical significance reporting, such as error bars or confidence intervals, which could strengthen the validity of the findings. This omission may raise concerns about the robustness of the results.
2.  Although the paper is generally well-written, some sections may benefit from additional elaboration, particularly in the explanation of the underlying mechanisms of the proposed pruning method. More detailed descriptions of the experimental setup and parameters could enhance reproducibility.
3. While the findings are significant, the paper could further explore the long-term implications of pruning on model performance, particularly regarding generation capabilities. A discussion on potential trade-offs between efficiency and performance in different contexts would add depth to the analysis.
4.  Although the paper introduces a novel framework, it could benefit from a more comprehensive comparison with existing expert pruning methods. Highlighting the unique advantages of EASY-EP over other approaches would strengthen the argument for its originality and effectiveness.

In summary, the paper presents a valuable contribution to the field of MoE models, with strengths in quality, clarity, significance, and originality. However, addressing the identified weaknesses could enhance its overall impact and acceptance in the research community.

---

> ### Author Rebuttal · Authors · 2025-07-31
>
> Thank you for your insightful suggestions!
> ## W1/Q1/L1: Statistical Significance Reporting
>
> We have provided the mean and standard deviation for computing error bars of DeepSeek-R1 on the AIME2024 dataset (32 samples), as shown in the table below. The results demonstrate that our method achieves better performance and comparable robustness compared to the full model baseline.
>
> | **Model** | **Mean Accuracy** | **Standard Deviation** |
> | --- | --- | --- |
> | Full Model | 0.7708 | 0.0232 |
> | Our Method | 0.7917 | 0.0289 |
>
> ## W2/Q2/L2: Reproducibility of Results
>
> We have provided all the experiment details in Appendix E and Section 5.1 to ensure the reproducibility of results.
>
> ## W3/Q3/L3: Impact of Expert Pruning on Long-term Generation Abilities
>
> In our paper, we analyze the gap in long-term generation abilities between the pruned reasoning and non-reasoning models. For reasoning models, the long-term generation abilities are easily harmed after pruning for baselines. However, our method can better preserve the long-term generation abilities with only half the experts in DeepSeek-R1
>
> ## W4: comprehensive comparison with existing expert pruning method
>
> In our paper, we conduct a comparison with existing methods, including expert pruning and merging. Our method can achieve significantly better performance than these methods, demonstrating the effectiveness of our method.
>
> In addition, there exist some perturbation-based methods that are not evaluated due to the huge computational costs. As analyzed in Appendix F, it will take over 10^7 for pruning one model, which is not useful for the large MoEs.
>
> ## Q4: Generalization Across Domains
>
> We analyze the generalization both in-domain and out-of-domain.
>
> **In-domain generalization:** To verify that our method is not only effective for a specific dataset, we conducted in-domain cross-dataset experiments in Section 3.2. As shown in Figure 2 (D), we found that within the mathematics domain, the overlap rate of important experts identified using different datasets (AIME-2023, AIME-2024, AIME-2025, HMMT-Feb 2025) exceeded 84%. This indicates that the experts identified by our method have strong in-domain generality and are not merely adapted to a specific calibration dataset.
>
> **Out-of-domain generalization:** We also evaluated the out-of-domain generalization capability. In Section 5.3.2 and Table 4, we show the performance on other unrelated domains after pruning with data from a specific domain. The experimental results show that the model exhibits a certain degree of generalization ability, especially between similar domains (e.g., mathematics/science, code/operating system agents). Meanwhile, we also point out that for multi-task scenarios, we recommend using a "mixed-domain pruning" strategy, as its cross-domain performance is superior to single-domain pruning.
>
> ## Q5/L5: Clarity and Structure of the Paper and Method
>
> We will improve the structure of our paper in the final version.
>
> ## L4: Potential Negative Social Impacts
>
>  Our work aims at the memory efficiency of large MoE models through a pruning technique. Thus, our method itself does not generate or exacerbate bias. Its impact is inherently linked to the societal impact of the base model being pruned, such as DeepSeek-R1.
>
> However, we fully agree that any technology that makes large models easier to deploy could indirectly amplify both the positive and negative societal impacts of these models themselves. In the final version, we will add a statement to the limitations section to clarify that while our method is technically neutral, its application should adhere to the responsible usage guidelines for the underlying large language models.

---

> > ### Author Response · Authors · 2025-08-05
> > **A friendly reminder that our discussion period ends soon**
> >
> > Dear reviewer,
> >
> > This is a kind reminder as the discussion period will end soon. We have made every effort to thoroughly address your comments in our responses, including conducting additional experiments and providing further analysis. If you have any remaining questions or concerns, please let us know at your earliest convenience.

---

> > ### Comment · Reviewer_mHRa · 2025-08-06
> >
> > I sincerely appreciate the author's response, which addressed most of my concerns. However, after careful consideration and taking into account the opinions of other reviewers, I have decided to uphold my initial score.

---

### Official Review · Reviewer_376Z · 2025-07-03

**Clarity:** 3
**Significance:** 2
**Originality:** 3
**Rating:** 4
**Confidence:** 4

**Summary:**

This paper investigates domain specialization in large Mixture-of-Experts (MoE) models. The authors observe that with just a few domain-specific examples, MoE models consistently activate a sparse and stable subset of experts ("few-shot expert localization").

Based on this observation, they develop a pruning framework, EASY-EP, that combines output-aware expert importance assessment with expert-level token contribution estimation to identify and retain only the most relevant experts for specific domains.

**Questions:**

1. The token contribution metric, $s_{t}^{l} = 1 - Sim(h_{t}^{l}, \tilde{h}_{t}^{l})$ , measures the representational change induced by the entire MoE module for a given token. Have the authors investigated how this change correlates with the properties of the activated experts? For example, does a larger change (lower similarity) correlate with higher variance in the gating scores for that token?

2. How does the method perform on other MoE architectures beyond DeepSeek models?

3. Can the method be extended to dynamic expert selection during inference?

**Ethical Concerns:**

["NO or VERY MINOR ethics concerns only"]

**Final Justification:**

In the discussion period, the authors have addressed my concerns raised about baseline choices and performance on more MoE models. I raise my score from 3 to 4 as a result.

**Limitations:**

The paper could benefit more by including a discussion of tasks that span multiple domains.

**Quality:**

2

**Strengths And Weaknesses:**

## Strengths
1. The discovery of "few-shot expert localization" is compelling and well-documented. The systematic analysis showing that only 5-25 demonstrations are sufficient to identify domain-specific experts (Figure 2C) is a valuable empirical finding that advances our understanding of MoE model behavior.

2. EASY-EP is well-designed with two complementary components. The method is computationally cheap, requiring only a single forward pass to calculate scores, making it practical even for models with hundreds of experts.

3. The paper is well-written

## Weaknesses
1. All experiments use DeepSeek variants. It's unclear whether the findings generalize to other MoE architectures (e.g., Moonlight-MoE, OLMoE, Mistral). This significantly limits the generalizability claims.

2. The comparison is mainly against simple baselines (random, frequency, gating scores). More sophisticated expert/model pruning methods from recent literature(e.g., MC-SMoE, Angular) could provide stronger baselines.

---

> ### Author Rebuttal · Authors · 2025-07-31
>
> Thank you for your insightful suggestions!
> ## W1&Q2: Experiments on more models
>
> We provide the experimental results on Qwen3-30B-A3B, the results are shown in the following Table. We can observe that for the MoE models without shared experts (which appear in DeepSeekMoEs), expert pruning is much harder. For example, the performance of other methods on AIME is very poor. On the contrary, our method can achieve over 80% performance on AIME with only 50% experts, further demonstrating the effectiveness and universality.
>
> | Qwen-30B-A3B | AIME24 | AIEM25 | HMMT | GPQA | LiveCodeBench |
> | --- | --- | --- | --- | --- | --- |
> | FULL | 80.42 | 70.83 | 50 | 68.59 | 62.8 |
> | Random | 0 | 0 | 0 | 23.23 | 0 |
> | frequency | 13.33 | 10 | 0 | 61.11 | 14 |
> | Gating Scores | 13.33 | 6.67 | 6.67 | 35.35 | 14 |
> | EASYEP | 72.92 | 61.25 | 38.75 | 63.13 | 46.3 |
>
> ## W2: Comparison with complex baselines
>
> As shown in our paper, we present M-SMoE (MC-SMoE is a further pruning version of M-SMoE, and its performance is worse than it). As shown in Table 2 in our paper and the following table, although the method is more complex, the performance of M-SMoE is significantly worse than our method, which demonstrates the effectiveness of our method.
>
> | Model | Method | Mix | \#E | AIME-24 | AIME-25 | FMMT | LiveCode | GPQA | USMLE | FinIQ | A-OS | Avg |
> | --- | --- | --- | --- | --- | --- | --- | --- | --- | --- | --- | --- | --- |
> | DeepSeek-R1 | Full | - | 256 | 77.08 | 66.67 | 44.38 | 63.32 | 70.91 | 92.66 | 82.1 | 40.51 | 67.20 |
> |  | M-SMoE | $\times$ | 64 | 0.00 | 0.00 | 0.00 | 0.00 | 12.12 | 0.00 | 0.00 | 0.00 | 1.52 |
> |  | EASY-EP | $\times$ | 64 | 72.81 | 55.10 | 38.02 | 42.51 | 67.47 | 26.63 | 33.90 | 27.26 | 45.22 |
> |  | M-SMoE | $\times$ | 128 | 5.33 | 6.00 | 3.33 | 25.75 | 24.75 | 52.63 | 39.60 | 19.44 | 22.10 |
> |  | EASY-EP | $\times$ | 128 | 79.17 | 68.33 | 45.31 | 61.11 | 70.12 | 91.67 | 78.80 | 37.92 | 66.55 |
> |  | M-SMoE | $\checkmark$ | 128 | 6.67 | 2.00 | 4.67 | 4.19 | 32.32 | 72.00 | 19.10 | 6.25 | 18.40 |
> |  | EASY-EP | $\checkmark$ | 128 | 75.94 | 61.98 | 42.50 | 57.63 | 70.36 | 91.20 | 57.95 | 34.17 | 61.47 |
> | DeepSeek-V3-0324 | Full | - | 256 | 55.73 | 47.71 | 28.75 | 48.50 | 66.87 | 87.51 | 64.22 | 33.33 | 54.08 |
> |  | M-SMoE | $\times$ | 64 | 16.67 | 13.33 | 3.33 | 1.20 | 22.22 | 12.18 | 47.00 | 21.52 | 17.18 |
> |  | EASY-EP | $\times$ | 64 | 53.12 | 41.56 | 28.85 | 27.99 | 57.35 | 84.57 | 72.50 | 27.55 | 49.19 |
> |  | M-SMoE | $\times$ | 128 | 48.00 | 38.67 | 28.67 | 30.53 | 55.82 | 86.72 | 66.60 | 33.33 | 48.54 |
> |  | EASY-EP | $\times$ | 128 | 55.21 | 46.88 | 31.56 | 46.71 | 65.25 | 86.72 | 63.58 | 37.08 | 54.12 |
> |  | M-SMoE | $\checkmark$ | 128 | 43.33 | 30.00 | 20.00 | 7.19 | 52.53 | 82.33 | 62.20 | 29.17 | 40.84 |
> |  | EASY-EP | $\checkmark$ | 128 | 57.81 | 46.56 | 33.33 | 40.72 | 64.95 | 85.00 | 72.26 | 38.74 | 54.92 |
>
> ## Q1: Relationship between similarities and variance in gating scores
>
> To investigate the relationship between the token-level representational change (measured by our similarity-based token contribution metric) and the properties of the activated experts, we computed the Pearson correlation between token similarity and the variance of the corresponding gating scores. The result was −0.3119, indicating a weak negative correlation. Though tokens with larger representational changes (i.e., lower similarity) tend to have slightly higher variance in their gating scores, this relationship is not strong.
>
> ## Q3: Potential for employment on dynamic expert selection during inference
>
> Yes, I think our method can also be employed in the task. We infer that the dynamic expert selection during inference aims to identify important experts in the MoE models via the initial tokens and to prune the MoE models for other tokens in the long generation. We analyze the top experts in the first 1000 tokens and all the tokens and observe that there are over 90% overlap ratios. Thus, we can dynamically prune experts according to the initial 1000 tokens for the long-generation tasks.

---

> > ### Comment · Reviewer_376Z · 2025-08-05
> >
> > Thank you for the comprehensive rebuttal and additional experiments. Your results on Qwen3-30B-A3B are indeed informative and show the challenges of expert pruning in models without shared experts.
> >
> > I would recommend conducting experiments on Moonlight-16B-A3B, as it provides a particularly valuable test case for the following reasons:
> >
> > - Architectural Similarity: Moonlight-16B-A3B shares the exact same MoE architecture as DeepSeek models (fine-grained experts + shared experts)
> >
> > - Controlled Comparison: Since Qwen3-30B-A3B lacks shared experts and shows much harder pruning behavior, Moonlight would serve as a better controlled experiment to test whether your "few-shot expert localization" phenomenon generalizes beyond the specific DeepSeek training regime.
> >
> > Could you consider including Moonlight-16B-A3B in your evaluation? This would provide a more nuanced understanding of when and why your method works effectively.

---

> > > ### Author Response · Authors · 2025-08-05
> > >
> > > Thank you for your insightful suggestions!
> > >
> > > We evaluate Moonlight-16B-A3B-Instruct with 64 experts per layer on the MathOAI dataset, since it is not a reasoning model. We prune the models to **16** experts (**25%** of the original expert numbers) per layer and compare EASY-EP with other baseline models. As shown in the following Table, our method can better preserve performances compared to other methods with a large compression ratio.
> > > | Method       | MathOAI | GSM8K |
> > > |--------------|---------|---------|
> > > | Full         | 65.2    |  78.49|
> > > | EASYEP       | 39.4    |33.21|
> > > | Frequency    | 8.6     |8.26|
> > > | Gating Score | 11.0    |11.14|

---

> > > > ### Comment · Reviewer_376Z · 2025-08-08
> > > >
> > > > The discussion has addressed my concerns, and I will raise my score. Good luck!

---

> ### Author Response · Authors · 2025-08-08
> **A friendly reminder that our discussion period ends in 1 day**
>
> Dear Reviewer,
>
> As the discussion period will conclude tomorrow, we would like to kindly remind you that we have provided detailed responses to your latest comments, including additional experiments and extended analyses. If there are any remaining questions or concerns, we would greatly appreciate it if you could share them at your earliest convenience.

---

### Official Review · Reviewer_kkGE · 2025-07-03

**Clarity:** 3
**Significance:** 3
**Originality:** 2
**Rating:** 4
**Confidence:** 4

**Summary:**

This paper investigates domain specific pruning of the experts in large SOTA MoE models (e.g., DeepSeek-R1 (671B)). The paper confirms that for such models, different expert specializes to different domain. Based on this confirmation, the paper designs a method for identifying domain-irrelevant experts for pruning them in order to reduce inference cost.

**Questions:**

1. How do you use the tokens of the pruned experts? Do you re-route them? Or, do you avoid computing the tokens?

**Ethical Concerns:**

["NO or VERY MINOR ethics concerns only"]

**Final Justification:**

The authors clarified a major part of the weaknesses. Therefore, I updated the rating.

**Limitations:**

Yes

**Quality:**

2

**Strengths And Weaknesses:**

Strengths:

1.	The empirical analysis provided in this paper to demonstrate the domain-specific specialization of experts in large-scale MoE models with many experts validates (probably for the first time in this paper) the hypothesis, and useful to MoE research community
2.	The proposed method of combining routing statistics with some expert’s output dependent metrics (output L2 norm and cosine similarity) outperforms some baselines provided in this paper
3.	The paper is well-written and easy to follow

Weakness:

1.	Although, the specific combination of metrics used in this paper for evaluating the experts’ importance is novel, most of these metrics are used before in literature.
2.	It’s not clear why the specific combination of metrics works better than the baselines provided in this paper, i.e., no theoretical justification has been provided. Specifically, for the case of using expert’s output L2 norm, equation (6) shows that larger the L2 norm of the output of an expert, larger the contribution of the expert in the overall L2 norm of the final output. But, why the contribution to final output’s L2 norm is a good measure?
3.	The comparison with existing baselines is inadequate. Although it has been mentioned that evaluating some of these baselines are computationally expensive (e.g., [1], [2]), still a comparison should be provided with those baselines to check that whether the proposed method performs worse than these methods. Specifically, those baselines should be included which doesn’t share any metric with the proposed combination of metrics (e.g., router-norm based method in [3]).

[1] Lu, Xudong, et al. "Not All Experts are Equal: Efficient Expert Pruning and Skipping for Mixture-of-Experts Large Language Models." Proceedings of the 62nd Annual Meeting of the Association for Computational Linguistics (Volume 1: Long Papers). 2024.

[2] Cao, Mingyu, et al. "Condense, Don't Just Prune: Enhancing Efficiency and Performance in MoE Layer Pruning." arXiv preprint arXiv:2412.00069 (2024).

[3] Chowdhury, Mohammed Nowaz Rabbani, et al. "A Provably Effective Method for Pruning Experts in Fine-tuned Sparse Mixture-of-Experts." Forty-first International Conference on Machine Learning.

---

> ### Author Rebuttal · Authors · 2025-07-31
>
> Thank you for your insightful suggestions!
> ## W1: Most of these metrics have been used before in literature
>
> We appreciate the reviewer’s observation that several of the metrics employed in our work have appeared in prior literature. We would like to clarify two key points regarding novelty and contribution.
>
> 1. For the issue of simplicity being mistaken for lack of novelty， our method’s simplicity is not due to insufficient innovation but is a deliberate design decision. For ultra-large MoE models such as DeepSeek-R1 (671B), overly complex pruning schemes are impractical. A lightweight scoring mechanism that requires only a single forward pass is crucial to ensure efficiency and deployability. Thus, simplicity is a necessity for practical impact rather than a limitation.
> 2. Regarding the reviewer’s request to clarify our core novelty beyond metric reuse, our primary contribution lies in uncovering the counterintuitive phenomenon of “few-shot expert localization”. We demonstrate, for the first time, that in ultra-large MoE models, identifying domain-specialized experts actually becomes *easier* with only a few demonstrations. This sheds light on the clear functional partitioning inside very large MoEs and enables effective domain-specific pruning.
> 3. In addition, compared to prior metric-based approaches, our method provides a stronger theoretical grounding. Specifically, we propose using the L2 norm of the gated expert output and the similarity between the input and output of MoE modules as the importance score, which can be interpreted as an approximation of the gradient-based contribution to the loss. Under this theoretical framework, our method serves as a principled proxy rather than a heuristic, ensuring that our pruning strategy aligns with the optimization objective. A detailed derivation is provided in our response to W2.
>
> ## W2: Why is the contribution to the final output’s L2 norm a good measure in theory?
>
> We provide a principled derivation from our optimization objective and gradient analysis, showing how the L2 norm is a good measure.
> ### Optimization Objective:
>
> Our pruning goal is to minimize the expected loss on a calibration dataset $D_{\text{calib}}$, under a constraint that only $M$ experts are retained per layer. Formally, we solve:
>
> $$
> \min_{m_l} \mathbb{E}^{(x,y)}  (\mathcal{L}(f(x; \{m_l\}), y)) , \quad \text{s.t. } \sum_{i=1}^N m_{il} = M
> $$
>
> Here, $m_{il} \in \{0, 1\}$ is a binary mask indicating whether expert $i$ in layer $l$ is retained.
>
> This is a combinatorial problem and is NP-hard. To solve it approximately, we adopt a greedy strategy that relies on estimating the importance of each expert.
>
> ### Gradient-Based Expert Importance:
>
> We approximate the change in loss when removing expert $j$ via a first-order Taylor expansion:
>
> $$
> \Delta \mathcal{L} \approx -\frac{\partial \mathcal{L}}{\partial m_{jl}}
> $$
>
> Applying the chain rule, we obtain:
>
> $$ \frac{\partial \mathcal{L}}{\partial m_{j,l}} = \sum_{t=1}^T \left( \frac{\partial \mathcal{L}}{\partial \tilde{h}{t,l}} \right)^T \left( g_{j,t,l} e_{j,t,l} \right) $$
>
> This shows that the importance of expert $j$ is proportional to the projection of the gradient of the loss onto the expert’s output.
>
> ### L2 Norm in Output-Aware Expert Importance Assessment:
>
> While we cannot compute the gradient $\partial \mathcal{L} / \partial \tilde{h}\_{t,l}$ during a forward-only pass, we **can** compute the expert’s contribution term $g\_{j,t,l} e\_{j,t,l}$.
>
> $$
> ( \frac{\partial \mathcal{L}}{\partial \tilde{h}\_{t,l}} )^\top  g\_{j,t,l} e\_{j,t,l} ) \leq \| \frac{\partial \mathcal{L}}{\partial \tilde{h}\_{t,l}} \|\_2 \cdot \| g\_{j,t,l} e\_{j,t,l} \|\_2
> $$
>
> This upper bound shows that **the L2 norm of $g_{j,t,l} e_{j,t,l}$** is a direct proxy for the expert’s contribution to the loss-sensitive direction in the model. Since the gradient component is unavailable during a forward pass, we approximate expert importance using this measurable term.
>
> Thus, **using the expert’s output L2 norm as part of the importance score is not ad hoc—it follows directly from a first-order gradient-based analysis of our objective.**
>
> ### Additional analysis of Expert-Level Token Contribution Estimation
> For expert-token contribution estimation, we need to compute $\partial \mathcal{L} / \partial \tilde{h}_{t,l}$. However, it is very costly to conduct gradient computation. Thus, we emply $1-Sim(\tilde h_t^l,h_t^l)$ as a approximation.
>
>
> ## W3: Comparison with perturbation-based methods and router-norm-based methods
>
> Due to the enormous parameter size of DeepSeek-R1 and the high complexity required by this method, it is difficult for us to implement it directly. As analyzed in our paper, NAEE and CD-MOE require 10^75 and 24,768 evaluations per layer, respectively. On a single H200 GPU, performing one round of pruning would require over 10^75 evaluations and approximately 9,976 days to complete (as described in Appendix E), which is unacceptable for current frameworks. Therefore, this method is, in any case, impractical to apply to models as large as DeepSeek-R1.
>
> For the router-norm-based method, we require the models before and after fine-tuning to evaluate the change in router. Thus, we select DeepSeek-V3-base and DeepSeek-R1 to select experts with the maximum L2 Norm of the difference of router weights and remove other experts. As shown in the following Table, this method can not achieve comparable performance to our method, which further demonstrates the effectiveness of our method.
> | Model | Method | Mix | \#E | AIME-24 | GPQA | LiveCodeBench |
> | --- | --- | --- | --- | --- | --- | --- |
> | DeepSeek-R1 | EASYEP | $\times$ | 128 | 79.17 | 70.12 | 61.11 |
> |  | EASYEP | $\checkmark$ | 128 | 75.94 | 70.36 | 57.63 |
> |  | Router-Norm | - | 128 | 16.67 | 50.00 | 20.96 |
> ## Q1: How do you use the tokens of the pruned experts?
>
> In our framework, tokens originally routed to the pruned experts are **not discarded**. Instead, they are **re-routed to the remaining experts** based on the gating mechanism. Specifically, after pruning, the router operates over the retained experts only, and each token is dynamically assigned to the most suitable expert(s) among this reduced set. This ensures that every token continues to receive expert processing without additional overhead.

---

> > ### Author Response · Authors · 2025-08-05
> > **A friendly reminder that our discussion period ends soon**
> >
> > Dear reviewer,
> >
> > This is a kind reminder as the discussion period will end soon. We have made every effort to thoroughly address your comments in our responses, including conducting additional experiments and providing further analysis. If you have any remaining questions or concerns, please let us know at your earliest convenience.

---

> > ### Comment · Reviewer_kkGE · 2025-08-05
> >
> > Thank you for your rebuttal. However, my concerns remain. I understand that, in this paper, the authors find domain-specific specialization of the experts and discovered the possibility of pruning domain-irrelevant experts. However, the major question is how to find the appropriate experts for pruning in an efficient way. The authors provide a few-shot localization solution without providing enough details of the reasoning behind using these metrics. In the rebuttal, the authors provide a theoretical reasoning behind using one of the metrics, namely the expert's output L2 norm. From the reasoning, we can see that the particular metric partially approximates the loss-sensitivity of the experts by using only the forward pass component, while the backward pass component is completely ignored. I understand that evaluating the backward pass component is computationally expensive for such large MoE models. However, incorporation of some cheap approximation methods, such as LoRA finetuning or low-bit approximation to approximate the backward-pass component (or even both components together), may surpass the results of the paper. Therefore, the fundamental question remains whether the proposed combination of metrics is the appropriate solution for the described scenario.
> >
> > The authors compared results with the router-norm-based method. However, the comparison is not appropriate as the DeepSeek-R1 is not trained only on the specified task.
> >
> > I asked the question on token re-routing to get an understanding of whether the described implementation can reduce the computation. However, as the tokens are being rerouted, we do not have any additional advantage of compute reduction. The only advantage is memory reduction. It would be interesting to see how much performance we can retain with token dropping using the proposed method. Also, if an expert is not domain-relevant, what is the purpose of retaining the corresponding tokens (note that there is still a residual version of the tokens)?
> >
> > I greatly appreciate the effort of the authors during the rebuttal.

---

> > > ### Author Response · Authors · 2025-08-06
> > >
> > > ## Whether the proposed combination of metrics is the appropriate solution for the described scenario.
> > > We appreciate the reviewer’s insightful comments. We would like to clarify why we  focus on a training-free, single-forward-pass solution, rather than adopting training-based methods such as LoRA finetuning or low-bit gradient approximations:
> > >
> > > 1. Practical constraints.
> > >     * First, there is a clear memory bottleneck: ultra-large MoE models (e.g., DeepSeek-R1 with 671B parameters) require over 1.3 TB of GPU memory, far beyond the 80 GB available on mainstream accelerators such as NVIDIA A800, making training impractical in most environments.
> > >     * Second, there are software and hardware limitations: in real-world applications, LLMs are typically deployed on inference-optimized devices that are designed for throughput rather than training. Such devices often lack support for training-critical toolkits and have environment restrictions that make gradient-based methods nontrivial to execute.
> > >     * Thirdly, there are application-level constraints. In many real-world use cases, such as personalized education, customer support, or task-specific assistants, the required expert configuration may change frequently depending on the domain, user intent, or task. In such dynamic settings, retraining or even lightweight finetuning for every configuration update becomes infeasible in practice.
> > >     * Finally, the great performance of our method has proved the effectiveness of our method.
> > >
> > > 2. Potential interference with the base model. The training trajectory of existing foundation models is largely opaque, covering pre-training corpora, domain-specific annealing data, and reinforcement learning stages. Directly finetuning with a small domain dataset to select experts risks disturbing the carefully optimized balance of the original model, potentially degrading its general-purpose capabilities.
> > >
> > > 3. Intrinsic issues of LoRA and low-bit approximations. Beyond the above, LoRA introduces architectural modifications (adding LoRA-A and LoRA-B modules) that increase operator and system-level overhead, which complicates deployment in already-optimized models. Low-bit approximations, if referring to quantization-aware training, demand even longer training time than standard fine-tuning, making them unsuitable in practice.
> > >
> > > ## More experiments about Router-based Method
> > >
> > > Thank you for pointing out that DeepSeek-R1 is obtained through multi-task finetuning based on V3. We would like to clarify the following:
> > > * First, our goal is to perform efficient pruning on existing models. For extremely large models such as DeepSeek-R1, additional fine-tuning is often impractical. Therefore, pruning methods that rely on finetuning — such as the router-norm-based approach — are generally infeasible in this setting.
> > > * Second, we compared our approach with both mix-domain pruning and fine-tuning-based methods. Similar to multi-task finetuning, our mix-domain pruning is not restricted to a single domain; however, it achieves significantly better performance.
> > > * Finally, given the enormous parameter size of R1, we conducted experiments on Qwen3-30B-A3B as a substitute. Using the mathematical problems from the STILL dataset [1] together with the model’s own outputs as training data, we performed SFT and then applied pruning based on router-norm changes. The results are shown below, where it can be observed that the router-norm-based method performs far worse than our approach.
> > >
> > > | Model | AIME24 | AIEM25 |
> > > |---|---|---|
> > > | FULL  | 80.42  | 70.83  |
> > > | Random | 0 | 0 |
> > > | Frequency    | 13.33  | 10 |
> > > | Gating Scores| 13.33  | 6.67 |
> > > | EASYEP       | 72.92  | 61.25 |
> > > | Router-norm  | 0| 3.33 |
> > > [1] Imitate, Explore, and Self-Improve: A Reproduction Report on Slow-thinking Reasoning Systems
> > > ## More experiments about expert-rerouted
> > >
> > > Thank you for your insightful suggestion!
> > > We experimented with the approach of **dropping tokens entirely instead of rerouting them**, as you proposed. Interestingly, this modification led to a slight improvement in performance. Specifically, we tested this variant (denoted as EASY-EP w/o reroute) on Qwen3-30B-A3B by dropping the tokens assigned to pruned experts, rather than rerouting them to others. The results are shown below. As can be seen, this variant achieves slightly better accuracy compared to the original EASY-EP (73.33 vs. 72.92 on AIME24 and 63.33 vs. 61.25 on AIME25). This observation further validates the soundness of expert pruning — not only are the pruned experts unnecessary, but the tokens they handle may also be redundant for the given domain. More importantly, it opens up the possibility of **further reducing computation** by skipping these tokens entirely, offering simultaneous gains in **both parameter reduction and inference efficiency** for MoE models.
> > >
> > > | Model | AIME24 | AIME25 |
> > > |---|---|---|
> > > | FULL | 80.42  | 70.83  |
> > > | EASYEP | 72.92  | 61.25  |
> > > | EASYEP w.o. reroute  | 73.33      | 63.33  |

---

> ### Author Response · Authors · 2025-08-08
> **A friendly reminder that our discussion period ends in 1 day**
>
> Dear Reviewer,
>
> As the discussion period will conclude tomorrow, we would like to kindly remind you that we have provided detailed responses to your latest comments, including additional experiments and extended analyses. If there are any remaining questions or concerns, we would greatly appreciate it if you could share them at your earliest convenience.

---

> ### Author Response · Authors · 2025-08-09
>
> Dear Reviewer,
>
> As the discussion period will soon conclude, we are writing to follow up on our submitted response. We hope the extended analyses and experiments have clarified our work, and we would be happy to address any final questions you may have.
>
> We appreciate your valuable feedback.

---

### Note · Authors · 2025-08-13

We express our sincere gratitude to all reviewers for their valuable feedback and insightful comments. We are glad to hear the acknowledgment from the reviewers:

- A valuable finding, termed as few-shot expert localization phenomenon, advances the understanding of MoE model behavior. (Reviewer 3TmE, mHRa, 376Z, kkGE)
- The method is computationally efficient, making it practical for large MoE models. (Reviewer mHRa, 376Z,3TmE)
- Highly competitive results on the most critical challenge, deploying large-scale MoE models (i.e., Deepseek-R1), across various benchmarks. (Reviewer 3TmE, mHRa).
- The paper is well-written. (Reviewer 376Z, kkGE)

We believe the experiments and analysis in the rebuttal have addressed all major concerns raised by the reviewers. Major concerns most reviewers shared are:

- Lack of theoritical support for the effectiveness of our method which are combination of several metrics.
  - We have analyzed our method from the perspectiveness of minimizing the loss. We transfer the expert pruning into a optimization problem. Through analysis of the gradient, We decompose the upper bound of the expert's gradient into two terms: output-aware expert importance and expert-level token contribution. For the former, we directly derive the product of the gating score and the L2 Norm of the expert's output. For the latter, we use the cosine similarity between the expert's input and output as a substitute to avoid expensive backpropagation.

- Only evaluated on Deepseek models and lack of some baselines and statistic analysis.
  - We have further evaluate our method on Qwen3-30B-A3B and Moonlight models. Experiments have demonstrated the effectiveness of our method.
  - We also compare our method with router-based methods and our method gains better performances on both domain-specific and mixed-domain pruning settings.
  - We report the statistical significance, where our method can achieve similar robustness compared to the original model.

We will polish our paper in the revised version. Specifically, we will make the following major changes:

- 1. We will add more experiments results with other models and baselines, as discussed in the rebuttal.
- 2. We will add theoritical analysis of our method in the paper.

Finally, we are encouraged that the reviewers recognized the potential of our work to advance the expert pruning of large MoE models. We will stimulate further discussion in the community.

---

### Decision · Program_Chairs · 2025-09-17

**Decision:**

Accept (poster)

**Comment:**

This paper presents that with just a few domain-specific examples, MoE models consistently activate a sparse and stable subset of experts ("few-shot expert localization"). Based on this observation, they develop a pruning framework, EASY-EP, that combines output-aware expert importance assessment with expert-level token contribution estimation to identify and retain only the most relevant experts for specific domains.

Strengths:
1. interesting findings.
2. simple and effective methods.

Weaknesses:
1. limited novelty (almost using existing similarity or norm-based measures)
2. the authors should add more baselines as suggested by the reviewers
3. the authors should test their methods on a more model families besides deepseek.